# Research on the promoting effect of servitization on export technological sophistication of manufacturing enterprises

Yuanhong Hu[1,2], Sheng Sun[3], Min Jiang[2], Yixin Dai[4]*

1 College of Economics & Management, Anhui Agricultural University, Anhui, Hefei, China, 2 College of Business, Shanghai University of Finance and Economics, Shanghai, China, 3 Research Department, Postal Savings Bank of China, Nanjing, Jiangsu, China, 4 College of Economics, Nanjing University, Nanjing, Jiangsu, China

* dg1602018@smail.nju.edu.cn

**Data Availability Statement:** The data underlying this study are owned by third-party sources and cannot be shared publicly by the authors. Future researchers can replicate the study findings in their

## Abstract

Based on multiple micro databases involving Chinese manufacturing enterprises and World Input-Output Database, this article investigates the impact of China's manufacturing servitization on export technological sophistication from 2000 to 2010. The results show that manufacturing servitization has an inverted U-shaped impact on export technological sophistication. From the perspective of heterogeneity at the enterprise level and industry level, manufacturing servitization has an inverted U-shaped impact on export technological sophistication for mixed trade enterprises, central and western located enterprises, domestic and foreign enterprises, and knowledge-intensive industries, the nonlinear impact is in the promotion range. Besides, manufacturing servitization with domestic and foreign service input source has an inverted U-shaped impact on export technological sophistication, manufacturing servitization with the domestic consumption-oriented service input source and foreign production-oriented service input source have a promoting effect. Servitization with financial industry and technical research and development service source has a promoting effect, while servitization with transportation service input source has an inverted U-shaped effect. Overall global value chain participation level and simple global value chain participation have a positive moderating effect on the impact, especially for enterprises with lower production efficiency. Mechanism analysis confirms that the "spillover" effect and "cost" effect are important channels for manufacturing servitization to promote export technological sophistication.

## Introduction

Manufacturing servitization refers to a complete "combination package" of products and services provided by manufacturing enterprises to customers, it is the integration of manufacturing that based on service and service-oriented manufacturing, and the integration of product economy that based on production and service economy that based on consumption [1]. There are many famous multinational companies throughout the world, such as IBM, General

entirely by directly obtaining the data from the following third parties and following the protocol in our Methods section of the manuscript. Chinese Enterprises Patent Database is from the Intellectual Property Press of the State Intellectual Property Office of China (+86-010-82000860; http://gjzscqcb.show.imosi.com/), China Customs Database is from the Survey Research Center, IAR, Shanghai University of Finance and Economics (+86-021-65902969; http://iarsrcen.shufe.edu.cn/), China Industrial Enterprises Database is from the National Bureau of Statistics of China, the contact (+86-10-68782000; http://www.stats.gov.cn/english/). WIOD is from World Input-Output Database (http://wiod.org/release16), export data is from the UN Comtrade database (https://comtrade.un.org/), and the rest of the variables are from World Bank WDI (https://databank.worldbank.org/reports.aspx?source=World-Development-Indicators/), which are freely available. The authors did not have any special access privileges that others would not have.

**Funding:** The authors acknowledge the support of the National Natural Science Foundation of China ("Human capital heterogeneity, innovation and productivity of producer services: impact and path" project, grant number 71773047, to YH; "Evolution mechanism and path selection of knowledge transfer and industrial innovation ecosystem in producer services from the perspective of symbiosis network" project, grant number 72073060, to YD); and the Social Science Foundation project of Jiangsu Province ("Analysis of the Influence Mechanism of FDI on Advanced Manufacturing Industry of Jiangsu Province under the Path of High-level Opening" project, grant number 18EYA004, to YD). The funders had no role in study design, data collection and analysis, decision to publish, or preparation of the manuscript. SS received funding in the form of salary from the Postal Savings Bank of China. The funder did not have any additional role in the study design, data collection and analysis, decision to publish, or preparation of the manuscript. The specific roles of this author are articulated in the 'author contributions' section.

**Competing interests:** The authors have declared that no competing interests exist.

Electric, Apple, and Nike that have realized the transformation towards production-oriented services in varying degrees. Compared with the western developed countries, the manufacturing servitization of China is still at a low level, and there are some prominent problems such as the lagging development of production-oriented services and the unreasonable industrial structure. Manufacturing servitization represents the general trend of development of manufacturing industries and its general direction of industrial transformation and upgrading. In the modern manufacturing supply chain, the majority of value-added is reflected and contained in the two ends of the "smile curve", that is, the front-end segment, e.g. design and R&D innovation, market research, consulting services, and the back-end segment, e.g. the third-party logistics, supply chain management optimization, and sales services. While the product segment in the middle is at the lowest position in the global value chain [2]. The extension for China's manufacturing segments to both front-end industries and back-end industries, and acceleration for the development of production-oriented services, are bound to contribute to the realization of high-end industry and the enhancement of international competitiveness of manufacturing enterprises.

With the comparative advantage of low production costs, China has gained the rapid growth of exports [3]. In recent years, with the gradual weakening and disappearance of demographic dividend and the increasingly rising cost in China, as well as the fact that the weak world demand and intensified trade conflicts between countries, the export from "quantity superiority" to "quality superiority" will become a new "breakthrough" to maintain the export stability and enhance international competitiveness. As an important aspect of export competitiveness, the issue concerning export technological sophistication has been paid more and more attention. Then, is manufacturing servitization enable to promote export technological sophistication of enterprises? The research of this article is of great theoretical and practical significance to clarify the relationship between them.

## Literature review and theoretical analysis

### Literature review

There are two types of literature related to this article. The first one is the research on the economic effect of servitization of the manufacturing industry. Most of this literature focus on the effect of manufacturing servitization on productivity and export. Most studies show that promoting the servitization of the manufacturing industry is an effective way to improve the competitiveness of enterprises [4, 5]. However, some scholars hold a more cautious view. [6] found the inverted U-shaped impact of manufacturing servitization on productivity based on the perspective of intermediate input, only when the intermediate service input is lower than the theoretical "optimal" critical value, the improvement of manufacturing servitization can promote the improvement of productivity. [7] used the sample of manufacturing servitization enterprises to find that the higher the degree of manufacturing servitization, the more likely the manufacturing enterprises to trigger the bankruptcy risk caused by the increase of internal risk. [8] found that find evidence of a servitization paradox, both S-firms and DS-firms show greater productivity than P-firms. The profitability of DS-firms is greater than that of P-firms, but the profitability of S-firms is lower than that of P-firms. [9] found that the interplay among service benefits, adjustment costs, and coordination costs results in a nonlinear relationship between servitization and business performance. A negative servitization–performance relationship is observed at low levels of servitization as adjustment costs would be dominant.

With the wide application of micro data, scholars begin to pay attention to the impact of manufacturing servitization on economic productivity and export at the enterprise level. [10] based on cluster analysis, they found four types of manufacturers have been identified

representing different stages of servitization. By multifactorial regressions, the impact of servitization on service networking has been investigated. Servitization is positively linked with increasing service networking activities of manufacturing companies. [5]'s empirical results also revealed a positive yet non-linear relationship between the scale of service activities and profitability: while initial levels of servicing result in a steep increase in profitability, a period of relative decline are observed before the positive relationship between the scale of services and profitability re-emerges. Besides, other scholars have carried out similar research and achieved fruitful results [11–13].

The second one is about the impact of manufacturing servitization on the international competitiveness of the industry. [14] confirmed a positive servitization-performance relationship. Besides, the results reveal that the observed servitization-performance relationship is influenced by the operationalization of constructs (servitization and performance) and control variables (industry and region). [6] found partial support for the moderating effects of the service supply network, which are characterized by tie strength and structural holes. Specifically, we show that stronger ties between a manufacturer and major service suppliers intensify the U-shaped servitization-performance relationship. Other scholars have also contributed to this line of research [15, 16].

As an important aspect of industrial competitiveness, export technological sophistication has also attracted the attention of academia in recent years. [17] based on measurement of export technological complexity and input servitization level in the manufacturing industry of China from 2006 to 2015, found the effects of onshore and offshore input servitization on export technological complexity of manufacturing industry were discussed based on the panel data model by combining industrial heterogeneity. Results demonstrated that offshore input servitization has significant positive effects on the export technological sophistication of the manufacturing industry and onshore input servitization can inhibit export technological complexity slightly. [18] investigated the impact of service outsourcing on China's export sophistication in manufacturing sectors presents a U-shaped relationship that first declines and then rises.

Through summarizing the existing literature, it is not difficult to find that although there is a research literature on the impact of manufacturing servitization on export technological sophistication. However, there are still many deficiencies. This article attempts to make a marginal contribution in the following three aspects, to make up for the shortcomings of the existing research. First, different from the conclusion that manufacturing servitization has a linear impact on export technological sophistication, this article finds that manufacturing servitization has an inverted U-shaped nonlinear impact on export technological sophistication, and further finds that the average value of manufacturing servitization is on the left side of the inflection point. In other words, the overall impact is in the promotion range. Second, different from the existing research on export technological sophistication at the country or industry level, this article measures the export technological sophistication index at the enterprise level on the foundation of [19]. Besides, considering the problem that it is impossible to accurately measure the sophistication index based on the total exports under the ubiquitous circumstance that intra-product trade and production specialization, this article adopts the export domestic value-added of a product instead of the exports as the weight basis. The new method can calculate the export technological sophistication index of enterprises more accurately in comparison with a traditional method. Thirdly, this article studies the impact of different sources of service input on the export technological sophistication of manufacturing enterprises.

## The mechanism of impact analysis

On the issue of how the integration of service elements into manufacturing export, in the process of servitization, affects the international competitiveness of the manufacturing industry, scholars mainly discuss it from two aspects. One is the "spillover" argument, which holds that manufacturing servitization will greatly improve the value-added and competitiveness of the manufacturing industry through its knowledge capital, technical capital, and human capital [20]. Besides, manufacturing enterprises use information technology such as the Internet of Things and Big Data Network to provide differentiated products for the market through customized services such as customer experience and online design, thus improving industrial competitiveness [21]. Another is the "cost" argument, which points out that manufacturing servitization has the advantages of specialization possessed by the production-oriented service industry, which makes production more efficient through economies of scale, thus reducing production costs and transaction costs [22, 23]. Besides, manufacturing servitization improves the efficiency of supply chain management through new technologies. The emerging technology, e.g. intelligent logistics realizes the management process optimization and market response acceleration of enterprises, to enhance industrial competitiveness [24, 25]. However, does manufacturing servitization only have a positive effect on the export technological sophistication of enterprises through the "spillover" effect and "cost" effect, as pointed out in the existing literature? Through theoretical analysis, this article attempts to put forward different views.

**The "spillover" effect mechanism.**   Manufacturing enterprises are the main entity of technological innovation and play a leading role in the development and change of industrial production technology. However, due to the limitations of R&D and production, high-end service input based on knowledge and technology is an important way for enterprises to form differentiated competitive advantage [26]. China's manufacturing industry as a whole is still mainly engaged in simple and repetitive processing and assembly business. The processing trade enterprises have the characteristics of "large imports and large exports, two ends outside", but there is a common absence of core technology research and development and high-end production technology. The improvement of manufacturing servitization will produce a "spillover" effect by deepening the integration of the high-end service industry into the manufacturing industry and promote the human capital and technical experience of enterprises. Hence, the promotion of the development of new products with innovative notions and upgrading of the quality and differentiation of export products will accelerate the improvement of R&D innovation level and export technology content of enterprises. However, due to the lack of knowledge assets reserve and the low efficiency of service input allocation management of processing trade enterprises, after manufacturing servitization reaches a certain level, it will also harm the R&D innovation vitality and space of enterprises. For ordinary trade enterprises, different market strategies are adopted for different customer groups through the input of service elements such as distribution, feasibility study, market consultation, etc., to promote enterprise technology research and development and enhance export technology content [27]. However, with the service input of the manufacturing industry occupying a more and more important position in the total factor input, the increase of the funds required for design, legal consultation, and other services also takes up R&D funds to a certain extent, which has an "extrusion" effect on R&D innovation activities.

**The "cost" effect mechanism.**   In the initial stage of improving service intermediate input, manufacturing service can directly reduce production costs by outsourcing higher efficiency and better quality of specialized production services. With the increase of service intermediate input, the advantages of high-end factor input are capable to be brought into full play, and the

"technological spillover" effect produced by knowledge capital and human capital can promote R&D innovation and export competitiveness of enterprises. [28] also pointed out that enterprises outsource their non-core segments to service enterprises, which helps to reduce additional costs and focus on their core segment production. However, since processing trade enterprises have been paying more attention to the expansion of tangible assets and production outputs for a long time, therefore the intangible assets of those kinds of enterprises such as production technology and human capital of enterprises are relatively weak. As a result, a situation that the most production-oriented services stay in the low-end service industry [29]. When manufacturing servitization exceeds a certain critical value, a continuous and blind increasing service input may lead to a sharp increase of the operating cost and management complexity of enterprises [30]. The increase in servitization is not conducive to the improvement of export technology content, which is known as the "service dilemma" [31]. Besides, when the servitization of the manufacturing industry is at a high level, the marginal promotion effect will gradually decrease with the increase of intermediate input per unit service. However, the excessive dependence caused by the long-term use of foreign service intermediate investment will increase the burden and pressure of high-end service payment to a certain extent, which is not conducive to the enhancement of export technology content.

Some scholars pointed out that there are significant differences between ordinary trade enterprises and processing trade enterprises in the form of production organization, and they need to undertake all the value-added segments of export products from R&D and design to production and sales. Therefore, the use of service intermediate input will help to promote the international competitiveness of enterprises [32]. This article holds different views on the argument. The ordinary trade enterprises also have the situation of service input outsourcing, and they are not necessarily involved in the total value-added segments in the process of production and export. The servitization of the manufacturing industry may have the same "dual-direction" as the processing trade enterprises in the export competitiveness effect. Besides, for a long time, ordinary trade enterprises have used more service intermediate investment in many product segments, which may make the marginal "spillover" effect caused by the improvement of servitization of the manufacturing industry no longer obvious.

## Research design

### Econometrical model specification

Econometrics is a discipline that uses mathematical and statistical methods to determine specific quantitative relationships in economic relationships. Econometrics measures and validates actual statistics of economic relationships and provides quantitative information for the qualitative account of economic theory on the dependence of economic variables to predict the future and provide a scientific basis for economic planning and determination of economic policy. In the subsequent study, we construct models based on econometric principles. To investigate the impact of manufacturing servitization on the export technological sophistication of enterprises, the benchmark econometrical model is specified as follows.

$$DTSI_{fit} = \alpha_0 + \beta_1 SR_{it} + \beta_2 SRSQ_{it} + \gamma X + \kappa_i + \mu_p + \eta_t + \varepsilon_{fit} \tag{1}$$

Where the subscript $f$, $i$, $p$ and $t$ denote the enterprise, industry, province, and year, respectively. $DTSI_{fit}$ denotes export technological sophistication of enterprises, as one of the important indicators of export performance, it measures the level of export product quality of enterprises in a narrow sense and the export competitiveness of enterprises in a broad sense. $SR_{it}$ and $SRSQ_{it}$ denote servitization and its square term, respectively. $X$ denotes control variables. Based on existing literature, this article selects the following variables [33–37]. $AGE_{it}$, the

enterprise age, which is measured by the difference between each year and the year when the enterprise was established. $SALE_{it}$, the sales of enterprises, which is measured by the annual total sales of industrial products. $FS_{it}$, the enterprise size, which is measured by the total assets scale of industrial products. $TFP_{it}$, the total factor productivity of enterprises, which is calculated by the LP method proposed by [38]. $HHI_{it}$, the industrial concentration, which is measured by a Herfindal index formula $HHI_{it} = \sum_{s \in Ind_j} (SALE_{ist}/SALE_{st})^2 = \sum_{s \in Ind_j} MS_{ist}^2$. In this formula, the sum of the squares of the share of sales of each industry in the total sales of a region constitutes the degree of industrial agglomeration in that region. When the proportion of a certain industry is larger, the final $HHI_{it}$ value will also be larger. $SOE_{it}$, the ownership of state-owned enterprises, which is measured by a dummy variable, and the state-owned ownership is set to 1, otherwise, it set to 0. $SUB_{it}$, the subsidy received by the enterprise, which is measured by the total amount of subsidy received by the enterprise in the current year. $MARKET_{pt}$, which is measured by the marketization index provided by [39], derived from a report titled The Relative Process of Marketization Index in Different Regions of China in 2013. For estimated parameters, $\alpha_0$ denotes the intercept term, or constant term is used to fit the model and has no practical significance. $\beta_1$ and $\beta_2$ denote the estimated parameters of the primary and secondary terms of the core variable manufacturing servitization, respectively. $\gamma$ denotes estimated parameters representing the set of control variables. For the estimated parameters of the fixed effects, $\kappa_i$, $\mu_p$ and $\eta_t$ denote the industry level, provincial level, and year level fixed effect, respectively. $\varepsilon_{fit}$ denotes the stochastic error term. Besides, to alleviate the influence of heteroscedasticity, the related variables except the dummy variable were treated by logarithm.

## Indicator measure

**Manufacturing servitization.** The servitization of the manufacturing industry is usually measured by the proportion of production-oriented service sector input in the total output of the downstream manufacturing industry. Based on the input-output analysis method [40], the direct consumption coefficient matrix A and the complete consumption coefficient matrix B are calculated by using the World Input-Output Database (WIOD). The direct consumption coefficient matrix measures the input of each service sector directly consumed by the manufacturing industry, while the complete consumption coefficient matrix measures the total consumption of each service sector in the unit output process of manufacturing sectors. Based on the research of [41] and [42], this article uses the input-output data of 56 sectors from 2000 to 2010 provided by WIOD in the 2016 edition and uses the complete consumption coefficient matrix to measure the servitization of China's manufacturing industry. The specific calculation formula is as follows.

$$SR_{ci} = SR_{ci}^d + SR_{ci}^f = \sum_s^S V_{ci} B_{cc}^{si} E_{ci} + \sum_{m \neq c} \sum_s^S V_{mi} B_{mc}^{si} E_{ci} \tag{2}$$

Where the subscript $c$, $i$, $m$ and $s$ denote the domestic country, manufacturing industry, foreign country, and production-oriented service industry, respectively. The superscript $d$, $f$ and $S$ denote the domestic service input source, foreign service input source, and production-oriented service industries consortium, respectively. $SR_{ci}$, $SR_{ci}^d$ and $SR_{ci}^f$ denote manufacturing servitization, the proportion of domestic service value-added contained in manufacturing export, and the proportion of foreign service value-added contained in manufacturing export, respectively.

After calculation, we present the trend of servitization level during 2000–2010 according to the 17 industries classified by WIOD in **Fig 1**. The level of manufacturing servitization in most industries shows a gradual decline.

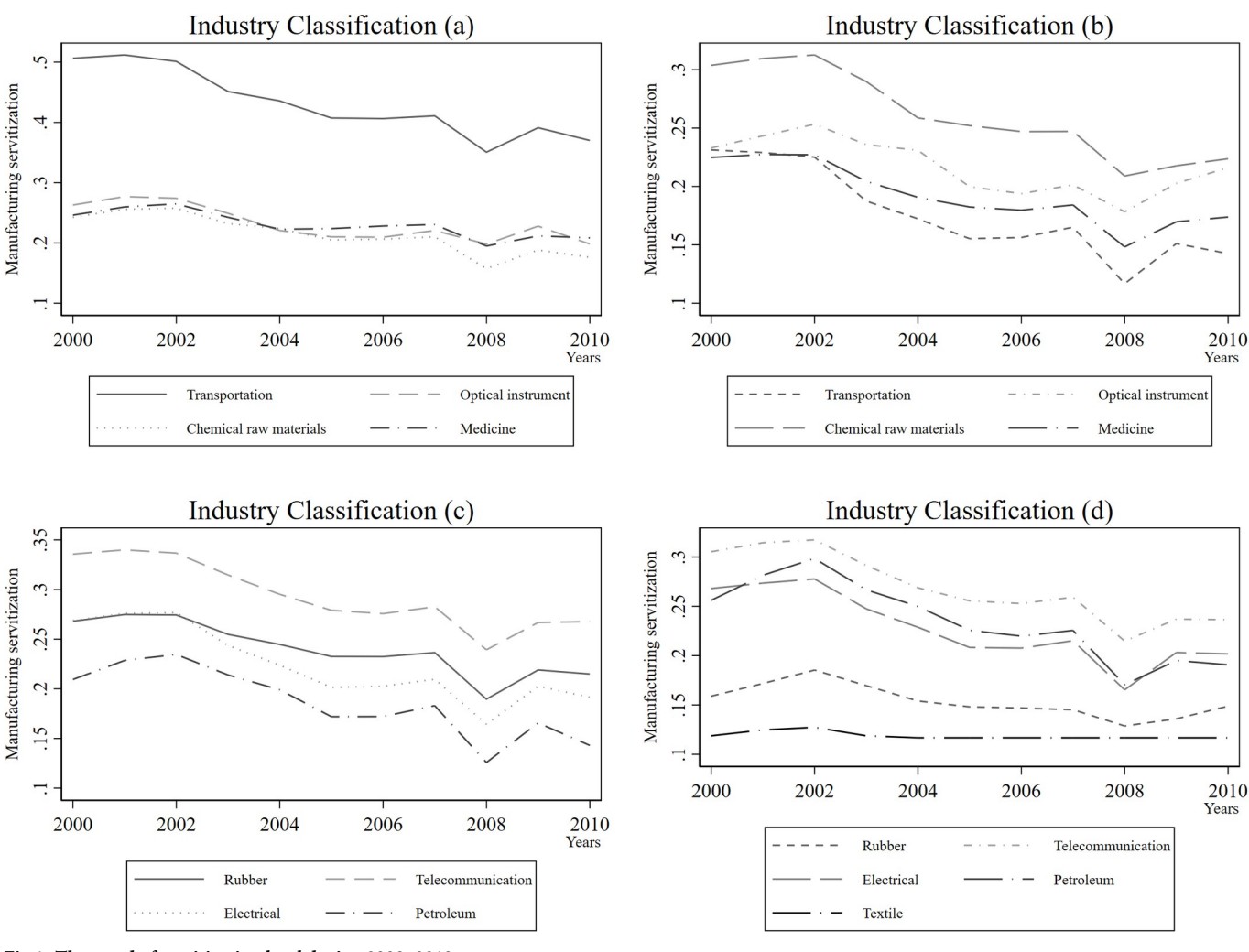

**Fig 1. The trend of servitization level during 2000–2010.**

**Export technological sophistication at the enterprise level.** With the refinement of international market fragmentation and the deep change of trade pattern, the export technological sophistication index calculated by traditional trade statistics method can not truly and accurately measure the technology content of export. Therefore, based on the research of [19], this article uses the improved two-step method of [43] to measure the export technological sophistication index. In the first step, the export technological sophistication index at the product level is calculated as follows.

$$PRODY_{pt} = \sum_j \frac{E_{pjt}/E_{jt}}{E_{pt}/E_{wt}} \cdot Y_{jt} \qquad (3)$$

Where $p$ denotes HS (Harmonized Standard) code 6-digit product, $j$ denotes the country, $E_{pj}$ denotes the exports of product $p$ in the country $j$, $E_j$ denotes the exports of the country $j$, $E_p$ denotes the total exports of product $p$ throughout the world, $E_w$ denotes the total exports of all products throughout the world, $Y_j$ denotes the per capita gross domestic product of a country $j$. The weight, $\Sigma_j[(E_{pj}/E_j)/(E_p/E_w)]$, is also called the revealed comparative advantage index.

In the second step, for consideration of robustness, this article adjusts the quality of export technological sophistication calculated in the first step by referring to the product relative price index method of [44], thereby yield $PRODY_{pt}^*$. The export technological sophistication index at the enterprise level is calculated as follows.

$$DTSI_{it} = \sum_p \frac{DVA_{ipt}}{DVA_{it}} \cdot PRODY_{pt}^* \tag{4}$$

Where $DTSI_{it}$ denotes adjusted export technological sophistication of the enterprise, which is included in the product quality adjustment, $DVA_{ipt}$ denotes the domestic value-added of the enterprise $i$'s product $p$, and $DVA_{it}$ denotes the total domestic value-added of the enterprise $i$, which is calculated from $\sum_p DVA_{ipt}$.

The export domestic value-added ratio and export domestic value-added of processing trade enterprise, ordinary trade enterprise, and mixed trade enterprise are calculated as follows.

$$\begin{cases} DVAR_{ipt}^o = 1 - \dfrac{IMP_{ipt}^o + IMPK_{ipt}^o - D_{ipt}^o}{Y_{ipt}} - \delta_{id}^F + \delta_{id}^D \\[2mm] DVAR_{ipt}^m = 1 - \dfrac{IMP_{ipt}^m + IMPK_{ipt}^m - D_{ipt}^m}{Y_{ipt}} - \delta_{id}^F + \delta_{id}^D \\[2mm] DVAR_{ipt}^x = \sigma_m \cdot (1 - \dfrac{IMP_{ipt}^o + IMPK_{ipt}^o - D_{ipt}^o}{Y_{ipt}} - \delta_{id}^F + \delta_{id}^D) \\[2mm] \qquad + \sigma_o \cdot (1 - \dfrac{IMP_{ipt}^m + IMPK_{ipt}^m - D_{ipt}^m}{Y_{ipt}} - \delta_{id}^F + \delta_{id}^D) \end{cases} \tag{5}$$

$$\begin{cases} DVA_{ipt}^o = DVAR_{ipt}^o \cdot Y_{ipt} \\[1mm] DVA_{ipt}^m = DVAR_{ipt}^m \cdot Y_{ipt} \\[1mm] DVA_{ipt}^x = DVAR_{ipt}^x \cdot Y_{ipt} \end{cases} \tag{6}$$

Where the superscript $o$, $m$ and $x$ denote the ordinary trade, processing trade, and mixed trade pattern, respectively. $IMPK_{ipt}$ denotes the actual capital goods imports, $D_{ipt}$ denotes the accumulated depreciation amount of imported capital goods, $\delta_{id}^F$ and $\delta_{id}^D$ denotes the foreign export value-added ratio and return value-added ratio of countries, respectively. $\sigma_m$ and $\sigma_o$ denote the proportion of processing trade exports and ordinary trade exports, respectively. Since the existing data do not provide the domestic sales and export amount of the imported intermediate products of the ordinary trading enterprises, it is assumed that the distribution of the imported intermediate products of the ordinary trade enterprises on the export products is proportional to the ratio of the export amount to the sales amount of the enterprises. In other words, the corresponding index for export is calculated based on the ratio of the enterprise's export amount to the sales amount as the standard by referring to the practice of [45].

The weight $DVA_{ipt}/DVA_{it}$ is used to measure the proportion ratio of export domestic value-added of product $p$ to export domestic value-added of an enterprise. For the calculation of export domestic value-added of enterprise's products, this article improves on a basis of the practice of [45] and [46], taking into account the trade agents, capital goods import, indirect import of intermediate goods and other issues, and recalculates the domestic value-added index of enterprise exports. Specifically, to the first, this article uses the method mentioned above to calculate the actual imports of intermediate goods and capital goods. Second, wipe out both import enterprises and export enterprises. Third, use the country-industry

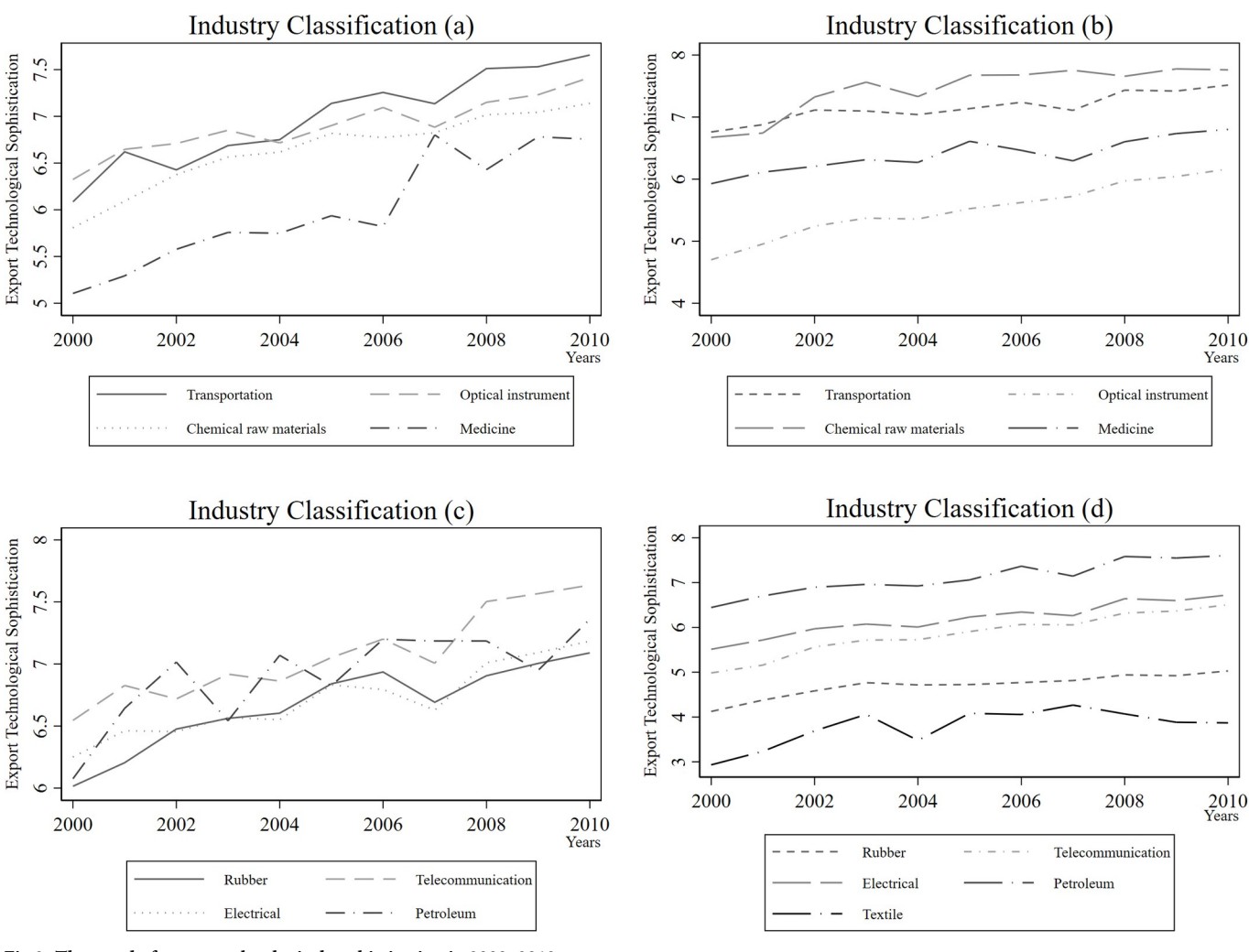

**Fig 2. The trend of export technological sophistication in 2000–2010.**

information in the WIOD to calculate the return value-added ratio and export foreign value-added ratio of China's manufacturing industries each year. Finally, calculate the export domestic value-added at the product level.

We report the trend of the calculated export technological sophistication index in a visualization according to the 17 industry criteria of WIOD in **Fig 2**. Some industries such as Transportation and Medicine have shown an upward trend in export technology complexity between 2000 and 2010, while others such as Rubber and Petroleum have maintained a flat trend.

**Data source and treatment.** The research sample period is from 2004 to 2010. Five sets of data are used in this article. The first one is China Industrial Enterprises Database, which is from the National Bureau of Statistics, covering the relevant data of all state-owned enterprises and non-state-owned enterprises above the designated size. The data of age, sales, liquid assets balance, and fixed investment are included. Given the problems of sample mismatch, missing indicators, and abnormal indicators in an industrial enterprise database, it is necessary to deal with them by referring to the practices of [33, 47]. Firstly, eliminate enterprises with less than 8 employees. Secondly, exclude enterprises with negative total assets, total fixed assets,

intermediate investment, and negative wages payable. Thirdly, eliminate enterprises established before 1949. The second one is China Customs Database, which is from the General Administration of Customs of China, covering all the 8-digit import and export data related to enterprise-level HS code. The import and export HS code, quantity, trade flows, trade patterns, and other information can be used to measure export technological sophistication, imports, and quality of intermediate products. Firstly, due to the problem of relying on trade agents in the import and export of Chinese enterprises, according to [48], the observation values containing the words "import and export", "economic and trade", "science and trade", "industry and trade", "trade" and "foreign economy" in the name of enterprises are eliminated. Secondly, the monthly data are added to the annual data, and the observation values with abnormal indicators are eliminated. The third one is WIOD. According to the input-output information between countries and sectors, this article uses the KWW model developed by [49] to decompose and calculate the return value-added ratio and export value-added ratio of industries of China. The fourth one is Chinese Enterprises Patent Database from the State Patent Office. The database contains patent applicants, application time, and other information, which can be used to match and calculate the number of patent applications. The fifth one is from the UN Comtrade Database. In this article, 5224 products from 141 countries in the world are selected to calculate the export technological sophistication of 6-digit HS code products. The rest variables are derived from World Bank WDI.

Throughout the manuscript, we use the statistical software StataMP16.1 to measure $DTSI_{it}$, $SR_{it}$ and other control variables. In addition, we use this software to perform panel fixed effects regressions on the models constructed in the whole manuscript, including mediating effects regressions to discriminate the impact mechanism and Heckman-Two Step method regressions to discriminate endogeneity.

The descriptive statistics and correlation of variables are presented in **Tables 1** and **2**.

## Empirical results and analysis

### Benchmark results and analysis

**Table 3** reports benchmark regression results from panel data spanning 2000–2010. Column (1) presents results that contain only coefficients of manufacturing servitization and its square term. It can be seen that the coefficients are 7.305 and -13.313, respectively, and the t value of the U-shaped test is 2.292, the statistic is significant at 5% level, which indicates that the impact

**Table 1. Descriptive statistics.**

| Variable | Observation | Mean | S.D. | Min | Max |
|---|---|---|---|---|---|
| DTSI | 1018000 | 6.432 | 4.142 | 0 | 10.11 |
| SR | 1018000 | 0.236 | 0.054 | 0.124 | 0.42 |
| AGE | 1018000 | 25.73 | 0.989 | 22.1 | 26.92 |
| MARKET | 989052 | 2.197 | 0.152 | 1.697 | 2.478 |
| UR | 935182 | 0.515 | 0.204 | 0.076 | 0.896 |
| HHI | 1018000 | 0.191 | 0.137 | 0.069 | 0.693 |
| SALE | 1018000 | 11.27 | 1.626 | 8.339 | 15.58 |
| FS | 1018000 | 11.19 | 1.674 | 7.677 | 15.52 |
| SOE | 1018000 | 0.047 | 0.212 | 0 | 1 |
| SUB | 1018000 | 0.001 | 0.003 | 0 | 0.025 |
| RD | 1018000 | 0.288 | 0.806 | 0 | 4.043 |
| COST | 1018000 | 11.12 | 1.651 | 8.109 | 15.87 |

**Table 2. Correlation matrix.**

|  | DTSI | SR | AGE | SALE | FS | SOE | SUB | MARKET | HHI | UR | RD | COST |
|---|---|---|---|---|---|---|---|---|---|---|---|---|
| DTSI | 1.000 |  |  |  |  |  |  |  |  |  |  |  |
| SR | 0.012 | 1.000 |  |  |  |  |  |  |  |  |  |  |
| AGE | 0.069 | -0.321 | 1.000 |  |  |  |  |  |  |  |  |  |
| SALE | 0.153 | -0.023 | 0.088 | 1.000 |  |  |  |  |  |  |  |  |
| FS | 0.163 | 0.003 | 0.066 | 0.876 | 1.000 |  |  |  |  |  |  |  |
| SOE | 0.022 | 0.034 | -0.024 | 0.198 | 0.264 | 1.000 |  |  |  |  |  |  |
| SUB | 0.002 | 0.017 | -0.044 | 0.053 | 0.101 | 0.142 | 1.000 |  |  |  |  |  |
| MARKET | 0.052 | -0.054 | 0.104 | -0.011 | -0.040 | -0.150 | 0.003 | 1.000 |  |  |  |  |
| HHI | -0.290 | -0.119 | -0.001 | -0.255 | -0.273 | 0.012 | 0.025 | -0.016 | 1.000 |  |  |  |
| UR | 0.033 | 0.016 | 0.009 | 0.025 | 0.049 | 0.021 | 0.005 | 0.164 | -0.052 | 1.000 |  |  |
| RD | 0.128 | -0.032 | 0.099 | 0.344 | 0.350 | 0.203 | 0.062 | -0.044 | -0.103 | 0.037 | 1.000 |  |
| COST | 0.154 | -0.033 | 0.109 | 0.982 | 0.861 | 0.198 | 0.042 | -0.009 | -0.251 | 0.024 | 0.347 | 1.000 |

of manufacturing servitization on the export technological sophistication of enterprises has an inverted U-shaped feature. Columns (2) and (3) show results of adding regional and enterprise-level control variables, respectively. Both the direction of the coefficient and the significance of U-shaped test results show that manufacturing servitization has an inverted U-shaped nonlinear effect on the export technological sophistication of enterprises. Column (4) presents results containing the control variables at the enterprise and regional levels. The coefficients of manufacturing servitization and its square term are 8.303 and -14.746, respectively. The t value of the U-shaped test is 3.704, which is statistically significant at the 1% level. Results are consistent with the conclusion of column (1), which shows that the impact of manufacturing servitization on export technological sophistication has a stable inverted U-shaped feature. Further analysis shows that the inflection point of column (4) is 0.282, while the overall average level of manufacturing servitization is at the level of 0.236, which is on the left side of the inflection point. It indicates that the impact of manufacturing servitization on export technological sophistication is in the promotion range, indicating that the improvement of manufacturing servitization will promote the improvement of export technological sophistication of manufacturing enterprises. The comparison between the mean value of column (1)-(3) and the inflection point also shows that it is consistent with the conclusion of column (4).

For the results of the control variables, we can see that the enterprise age, enterprise productivity, and industrial concentration coefficient are significantly negative. The reasonable explanation is that the enterprises found earlier tend to pay less attention to the content of export technology. The enterprises with higher productivity bear lower average production costs and lack the internal power to enhance the export technology content to maintain competitiveness. For enterprises with higher industry concentration, will obtain excess profits under monopoly position, and have no external pressure to enhance the content of export technology. Sales scale and enterprise-scale coefficient are significantly positive. The explanation is that the enterprises with larger market share and larger scale not only have abundant funds to engage in R&D innovation to improve export technology level, also need to improve export competitiveness through quality upgrading, because with the expansion of production outputs, but the marginal contribution of its "scale economy" effect to enhance competitiveness is also definitely decreased continuously.

**Table 3. The results of benchmark regression.**

| Variable | (1) DTSI | (2) DTSI | (3) DTSI | (4) DTSI |
|---|---|---|---|---|
| SR | 7.124*** | 7.787*** | 6.774*** | 7.375*** |
|  | (1.37) | (1.04) | (1.37) | (1.07) |
| SRSQ | -13.230*** | -13.545*** | -13.023*** | -13.380*** |
|  | (3.33) | (2.35) | (3.14) | (2.36) |
| AGE |  |  | -0.081*** | -0.106*** |
|  |  |  | (0.02) | (0.02) |
| SALE |  |  | 0.065 | 0.022 |
|  |  |  | (0.05) | (0.05) |
| FS |  |  | 0.307*** | 0.145*** |
|  |  |  | (0.04) | (0.05) |
| SOE |  |  | -0.070 | 0.367*** |
|  |  |  | (0.09) | (0.08) |
| SUB |  |  | -12.013*** | -2.002 |
|  |  |  | (4.36) | (4.26) |
| MARKET |  | -0.445 |  | -0.421 |
|  |  | (0.37) |  | (0.36) |
| HHI |  | -8.467*** |  | -7.966*** |
|  |  | (0.30) |  | (0.29) |
| UR |  | 0.078 |  | 0.140 |
|  |  | (0.20) |  | (0.20) |
| Intercept | 5.429*** | 7.877*** | 3.457*** | 8.631*** |
|  | (0.16) | (0.73) | (0.87) | (1.16) |
| Industry fixed effect | Yes | Yes | Yes | Yes |
| Province fixed effect | Yes | Yes | Yes | Yes |
| Year fixed effect | Yes | Yes | Yes | Yes |
| U-shaped test t value | 2.364*** | 3.211*** | 2.637*** | 3.400*** |
| P value | 0.009 | 0.001 | 0.004 | 0.000 |
| Inflection point | 0.269 | 0.287 | 0.260 | 0.276 |
| Mean | 0.236 | 0.236 | 0.236 | 0.236 |
| F statistics | 14.68 | 239.9 | 56.40 | 226.8 |
| Adjusted R-squared | 0.042 | 0.115 | 0.060 | 0.119 |
| Observation | 958944 | 931416 | 958944 | 931416 |

Note: *, ** and ** is statistically significant at 10%, 5% and 1%, respectively. The models controlled for industry, province and year fixed effects. The robust standard errors of clustering at city level are presented in parentheses. F statistic indicates that the model has joint significance.

## Mediation effect analysis

The results show that the impact of manufacturing servitization on export technological sophistication has an inverted U-shaped nonlinearity, and it is in the promotion range. Then, what kind of internal mechanism does manufacturing servitization promote export technological sophistication? The investigation of the internal mechanism is helpful to understand the internal relationship between manufacturing servitization and export technological sophistication. Therefore, this article introduces mediating effect models to test the transmission mechanism behind the impact. According to the theoretical analysis in the second part of this article, two variables, R&D innovation, and production cost, are selected as mediating variables, and the test method of mediating effect proposed by [50] and [51] is used as a reference to develop

a mediating effect test model.

$$DTSI_{fit} = \beta_0 + c_1 SR_{it} + c_2 SRSQ_{it} + \gamma X + \kappa_i + \mu_p + \eta_t + \varepsilon_{fit} \tag{7}$$

$$RD_{fit} = \beta_1 + a_1 SR_{it} + a'_1 SRSQ_{it} + \gamma X + \kappa_i + \mu_p + \eta_t + \varepsilon_{fit} \tag{8}$$

$$COST_{fit} = \beta_2 + a_2 SR_{it} + a'_2 SRSQ_{it} + \gamma X + \kappa_i + \mu_p + \eta_t + \varepsilon_{fit} \tag{9}$$

$$DTSI_{fit} = \beta_3 + c'_1 SR_{it} + c'_2 SRSQ_{it} + b_1 RD_{fit} + b_2 COST_{fit} + \gamma X + \kappa_i + \mu_p + \eta_t + \varepsilon_{fit} \tag{10}$$

Where the subscript $f$, $i$, $p$ and $t$ denote the enterprise, industry, province, and year, respectively. $DTSI_{fit}$ denotes export technological sophistication of manufacturing enterprise. $SR_{it}$ denotes manufacturing servitization. The mediation variable $RD_{fit}$ denotes the R&D innovation of enterprises, which is measured by the number of patent authorizations of enterprises. Specifically, it is estimated regarding the practice of [52]. Based on the research of [53], the patent depreciation rate is set to 0.15, and the relevant patent data are from the China Enterprise Patent Database. The mediation variable $COST_{fit}$ denotes the production cost of an enterprise, which is obtained by adding the cost of the main business and the cost of sales. The control variables $X$ are consistent with the benchmark model, $\kappa_i$, $\mu_p$, $\eta_t$ and $\varepsilon_{fit}$ denote the industry level, provincial level, and year level fixed effects and random errors, respectively.

**Fig 3** depicts the variable relationships of mediating effects. For simplicity, the presence of quadratic terms of the servitization $SR$ is not considered. $ETS$, $SR$, $RD$ and $COST$ Eqs (7)–(10) and the corresponding estimated coefficients are presented in **Fig 3**.

**Table 4** reports the test results of the mediation on the impact of manufacturing servitization on export technological sophistication. Columns (1)-(4) are the test results of mediating effect of R&D innovation. The dependent variable of column (1) is R&D innovation, and the coefficient of manufacturing servitization is 0.190, which indicates that manufacturing servitization promotes R&D innovation of enterprises, but the coefficient does not pass the significance test. In column (3), the estimated coefficient of the mediation variable is significantly positive, indicating that R&D innovation has a significant role in promoting the export technological sophistication of enterprises. Columns (5)-(8) are the test results of the mediating effect of production cost. Where the dependent variable of column (5) is the production cost and the coefficient of manufacturing servitization variable is -0.061, and has 1% statistical significance, indicating that the manufacturing servitization significantly inhibits the rise of enterprise production cost. In column (8), the estimated coefficient of the mediation variable is significantly negative, which indicates that the production cost has a significant inhibitory effect on export technological sophistication.

The test for the coefficient of interaction is the core of mediating effect test. However, in practice, it often occurs that the coefficient of interaction is significant but the sequential test is not significant [54–56]. Therefore, in the sequential test, if at least one of the coefficients $a$ and $b$ is not significant, it is necessary to further test the coefficient of interaction to determine whether there is mediating effect. Generally speaking, a Sobel method [57] is the most widely used method in the test of coefficient of interaction. This method tests the statistics $z = \hat{a}\hat{b}/S(\hat{a}\hat{b})$, where $\hat{a}$ and $\hat{b}$ is the estimation of the coefficient $a$ and $b$, and $S(\hat{a}\hat{b})$ is the standard error of $\hat{a}$ and $\hat{b}$. Empirical studies show that the Sobel method is more powerful than the sequential test [51, 55]. If the results of the coefficient of interaction test are statistically significant, it indicates that the mediating effect exists. Using the Sobel method, the test of both mediation variables are 6.732 and 11.840, respectively, which are significant at a 1% level. Besides,

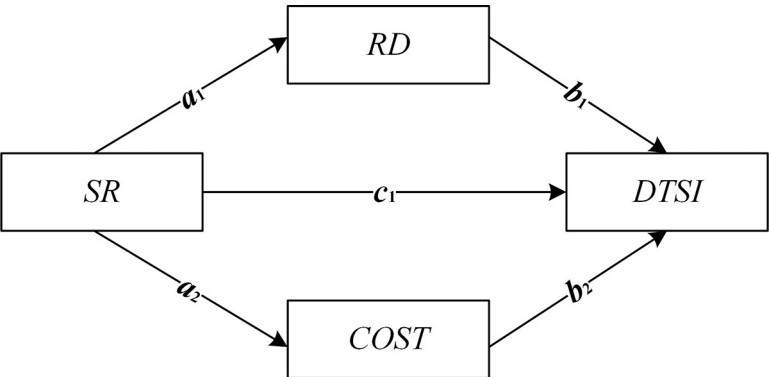

**Fig 3. The variable relationships of mediating effects.**

in columns (4) and column (8) with the mediation variables of enterprise R&D innovation and production cost, the estimation coefficient of manufacturing servitization is significantly lower than that of the benchmark model, both in value and significance. The above results fully show that R&D innovation and production cost are two possible mediation channels for manufacturing servitization to affect export technological sophistication. In column (9), the two mediation variables are added to the regression equation to estimate. It is found that the estimation coefficient of manufacturing servitization is still positive, but compared with the result of benchmark regression, its estimated coefficient is smaller and has not passed the significance test of 10%. However, the estimated coefficients of R&D innovation and production cost of mediation variables are consistent with the corresponding results of column (4) and

**Table 4. The results of mediation effect.**

| | (1) | (2) | (3) | (4) | (5) | (6) | (7) | (8) | (9) | (10) |
|---|---|---|---|---|---|---|---|---|---|---|
| Variable | RD | RD | DTSI | DTSI | COST | COST | DTSI | DTSI | DTSI | DTSI |
| SR | 0.190 | 0.718 | 0.638 | 4.501** | -0.061* | -0.085 | 0.483 | 7.942*** | 0.596 | 4.384** |
| | (0.32) | (0.93) | (0.90) | (2.18) | (0.03) | (0.07) | (0.56) | (1.07) | (0.90) | (2.20) |
| SRSQ | | -1.030 | | -7.532* | | 0.044 | | -13.864*** | | -7.386* |
| | | (1.77) | | (4.20) | | (0.12) | | (2.38) | | (4.24) |
| RD | | | 0.075* | 0.075* | | | | | 0.080** | 0.079** |
| | | | (0.04) | (0.04) | | | | | (0.04) | (0.04) |
| COST | | | | | | | 0.062 | 0.062 | -0.267** | -0.266** |
| | | | | | | | (0.10) | (0.10) | (0.13) | (0.13) |
| Intercept | -2.894** | -2.959** | 8.887*** | 8.410*** | -0.388*** | -0.385*** | 9.188*** | 8.230*** | 8.641*** | 8.174*** |
| | (1.23) | (1.21) | (2.18) | (2.26) | (0.10) | (0.10) | (1.19) | (1.15) | (2.17) | (2.26) |
| Control variables | Yes | Yes | Yes | Yes | Yes | Yes | Yes | Yes | Yes | Yes |
| Industry fixed effect | Yes | Yes | Yes | Yes | Yes | Yes | Yes | Yes | Yes | Yes |
| Province fixed effect | Yes | Yes | Yes | Yes | Yes | Yes | Yes | Yes | Yes | Yes |
| Year fixed effect | Yes | Yes | Yes | Yes | Yes | Yes | Yes | Yes | Yes | Yes |
| F statistics | 25.7 | 23.1 | 101.9 | 96.3 | 716.5 | 646.5 | 351.3 | 341.1 | 92.6 | 87.7 |
| Adjusted R-squared | 0.217 | 0.217 | 0.121 | 0.121 | 0.969 | 0.969 | 0.123 | 0.123 | 0.122 | 0.122 |
| Observation | 134108 | 134108 | 133384 | 133384 | 992412 | 992412 | 989052 | 989052 | 133384 | 133384 |

Note: *, ** and ** is statistically significant at 10%, 5% and 1%, respectively. The models controlled for industry, province and year fixed effects. The robust standard errors of clustering at city level are presented in parentheses. F statistic indicates that the model has joint significance.

column (8) in terms of coefficient sign and significance, which shows that manufacturing servitization indeed promotes export technological sophistication of enterprises through the two mediation channels of R&D innovation and production cost.

To determine the dominant role of the R&D innovation effect and production cost effect in the impact, this article uses the method of Wen et al. [51] to calculate the proportion of the two mediating effects $Pr_{RD} = a_1 b_1 / c'$ and $Pr_{COST} = a_2 b_2 / c'$ in the total effect. According to the estimation results of column (9), the proportion of R&D innovation effect and production cost effect is 0.026 and 0.053, respectively, and the latter is just twice as much as the former. The results show that manufacturing servitization promotes export technological sophistication through R&D innovation and production cost effect channels respectively, but the promoting effect of production cost effect is significantly greater than that of an R&D innovation effect, which also brings obvious enlightenment. Under the trend of manufacturing servitization, many traditional manufacturing enterprises in China have gradually shifted from production-oriented manufacturing to service-oriented manufacturing. However, domestic export enterprises, especially those engaged in a simple assembly and rough processing, are mainly engaged in the production and export of labor-intensive, capital-intensive, and resource-intensive products. Compared with taking advantage of the technology spillover effect brought by manufacturing servitization, domestic export enterprises can enhance their export technology content by increasing R&D innovation. It will reduce its production costs employing specialization of production, economies of scale, and efficiency improvement of supply chain management, to realize the effective improvement of its export technology level. Besides, columns (2), (4), (6), (8), and (10) report the corresponding econometric results by adding a square term of manufacturing servitization. The core results are correspondent with the results above.

## Heterogeneity results and analysis

**The results and analysis of heterogeneity in enterprise characteristics.** From the results of benchmark regression, it can be seen that manufacturing servitization has an inverted U-shaped nonlinear impact on the export technological sophistication of enterprises as a whole, and this impact may be heterogeneous in both the enterprise characteristics and the industry characteristics. This article starts from the perspective of heterogeneity of enterprise characteristics and classifies the sample enterprises according to the characteristics of enterprise trade pattern, region, and ownership to further investigate the heterogeneous impact of manufacturing servitization on export technological sophistication.

On the sample division of heterogeneity in the enterprise characteristics. This article divides the enterprise trade pattern into ordinary trade, processing trade and mixed trade enterprises based on the standard of export share (Mao & Xu, 2019). Enterprises in Beijing, Tianjin, Hebei, Shanghai, Jiangsu, Zhejiang, Fujian, Shandong, Guangdong and Hainan provinces are divided into eastern enterprises based on the standards of the National Bureau of Statistics, and enterprises in the rest areas are divided into central and western enterprises. The China-foreign cooperative enterprises, sino-foreign joint ventures and wholly foreign-owned enterprises are divided into foreign-funded enterprises, and other enterprises are divided into domestic enterprises. **Table 5** reports the results of heterogeneity based on trade patterns, regions, and ownership. Columns (1)-(3) are the results of trade pattern heterogeneity. The coefficients of manufacturing servitization and its square term in ordinary trade enterprises are not significant, which indicates that manufacturing servitization has no significant impact on the export technological sophistication of ordinary trading enterprises. The core variable coefficients of processing trade enterprises are statistically significant, but the t value of the U-shaped test is 0.789, which does not pass the significance test, which shows that manufacturing

**Table 5. The results of heterogeneity in enterprises characteristics.**

| | (1) | (2) | (3) | (4) | (5) | (6) | (7) |
|---|---|---|---|---|---|---|---|
| | Trade Pattern of Enterprises | | | Region of Enterprises | | Ownership of Enterprises | |
| Variable | Ordinary Trade | Processing Trade | Mixed Trade | East | Central-West | Domestic-Funded | Foreign-Funded |
| *SR* | 0.873 | 4.277** | 4.671*** | 6.778*** | 12.750*** | 16.828*** | 6.217*** |
| | (1.60) | (2.07) | (1.52) | (1.12) | (4.13) | (2.49) | (1.20) |
| *SRSQ* | -2.760 | -12.611*** | -7.249** | -12.099*** | -24.706*** | -29.472*** | -11.104*** |
| | (2.95) | (3.42) | (2.83) | (2.47) | (7.10) | (4.84) | (2.30) |
| Intercept | 13.123*** | 2.307 | 8.608*** | 8.831*** | 4.938*** | 8.883*** | 8.120*** |
| | (0.73) | (2.95) | (1.24) | (1.28) | (1.90) | (2.36) | (1.03) |
| Control variables | Yes | Yes | Yes | Yes | Yes | Yes | Yes |
| Industry fixed effect | Yes | Yes | Yes | Yes | Yes | Yes | Yes |
| Province fixed effect | Yes | Yes | Yes | Yes | Yes | Yes | Yes |
| Year fixed effect | Yes | Yes | Yes | Yes | Yes | Yes | Yes |
| U-shaped test t value | 0.195 | 0.826 | 1.195 | 2.804*** | 2.691*** | 3.912*** | 3.064*** |
| P value | 0.423 | 0.205 | 0.117 | 0.003 | 0.004 | 0.000 | 0.001 |
| Inflection point | 0.158 | 0.170 | 0.322 | 0.280 | 0.258 | 0.285 | 0.280 |
| Mean | 0.125 | 0.362 | 0.201 | 0.158 | 0.182 | 0.230 | 0.215 |
| F statistics | 21.9 | 57.9 | 224.2 | 246.4 | 57.2 | 175.3 | 306.1 |
| Adjusted R-squared | 0.069 | 0.097 | 0.227 | 0.117 | 0.121 | 0.128 | 0.121 |
| Observation | 201110 | 205116 | 420101 | 837647 | 93769 | 162477 | 768939 |

Note: *, ** and ** is statistically significant at 10%, 5% and 1%, respectively. The models controlled for industry, province and year fixed effects. The robust standard errors of clustering at city level are presented in parentheses. F statistic indicates that the model has joint significance.

servitization has only a significant positive linear impact on the export technological sophistication of processing trade enterprises. The core coefficients of mixed trade enterprises are 5.606 and -8.600, which are statistically significant. From the comparison of an inflection point and mean value, the mean value is on the left side of the inflection point, which indicates that manufacturing servitization has an inverted U-shaped nonlinear impact on the mixed trade enterprises and is in the promotion range. Columns (4)-(5) are the results of regional heterogeneity. The core coefficient and U-shaped test results show that manufacturing servitization has an inverted U-shaped nonlinear impact on the export technological sophistication of enterprises in the eastern and central-western regions. The mean values of each sample are located on the left side of the corresponding inflection point, indicating that their impacts are in the promotion range. In terms of inflection point, eastern enterprises are larger than central-western enterprises, which indicates that the promotion range of manufacturing servitization for the former is longer and the promotion space is larger. The explanation is that the eastern enterprises have a better business environment than the central-western enterprises, and their growth rate is faster. Therefore, manufacturing servitization plays a more obvious role in promoting the export technology content of eastern enterprises in the process of the high-end industry. Columns (6)-(7) are the results of ownership heterogeneity. The core coefficients and U-shaped test pass the significance test, and their mean values are on the left side of the inflection point, indicating that manufacturing servitization has an inverted U-shaped nonlinear impact on the export technological sophistication of domestic and foreign-funded enterprises, and is in the promotion range. From the comparison of inflection point, domestic enterprises are larger than foreign enterprises, which indicates that the promotion range of manufacturing servitization for the former is longer and the promotion effect is more obvious. There are two possible reasons. One is that the direct purpose of foreign-funded enterprises

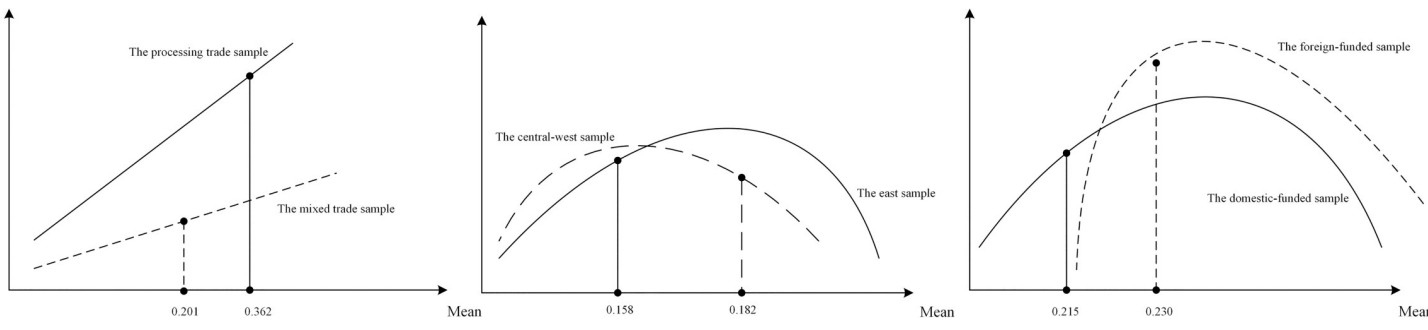

**Fig 4. Visualization of heterogeneous regression results.**

found by foreign multinational companies is to engage in the simple and repetitive offshore outsourcing business while neglecting the optimization of production structure and technical improvement. The other is that foreign-funded enterprises have a comparative disadvantage in localization compared with domestic enterprises. Therefore, the improvement of domestic manufacturing servitization is not enabling to bring a good "spillover" effect on the export technology level of foreign enterprises.

We further report the main findings of the heterogeneity analysis in a visual form in **Fig 4** as an aid to more visually represent the economic implications behind the findings.

## The robustness test

**The endogeneity concern.** For empirical research, endogeneity is an unavoidable issue. If the core independent variables are not strictly exogenous, the regression results will be biased and inconsistent. The core independent variable manufacturing servitization in this article, as an industry level indicator, has a certain degree of exogenous relative to the enterprise-level data. However, the manufacturing servitization of an enterprise has a potential correlation with the specific industry. Therefore, there may be a certain degree of reverse causality problem, that is, the endogeneity under the mutual impact of dependent variables and independent variables. The instrumental variables and two-stage least square method 2SLS are introduced to solve it. Besides, the data used in this article retain the manufacturing enterprises with export behavior, which may lead to the endogenous problem of sample selection bias, that is, the observation value only comes from the non-random finite individuals. Therefore, the Heckman two-step method [58] is used to test whether there is sample selection bias in the model. Finally, this article always controls the fixed effects of industry, province, and year in the regression, and uses the robust standard error of city-level clustering. Therefore, the endogeneity of missing variables, that is, the endogeneity problem caused by the coverage of unobservable factors that may potentially affect the core independent variables in the random error term of the model, may not be serious.

Columns (1)-(4) in **Table 6** are the results of the endogenous test. Where columns (1)-(2) report the results of Heckman's two-step estimation. The estimation coefficients of core variables in the first and second stage regression have passed the significance test, are it is consistent with the benchmark results, indicating that the core conclusion is still valid. The inverse mills ratio *IMR* is statistically significant at the 1% level, which indicates that there is a certain degree of selection bias in the benchmark regression, and it is reasonable to adopt the Heckman two-stage method to control sample selection bias.

Columns (3)-(5) report the results of the two-stage least squares (2*SLS*) estimation. First of all, in the relevant literature, most scholars choose lag one period value of the endogenous

**Table 6. The results of indicator robustness test.**

| Variable | (1) | (2) | (3) | (4) | (5) | (6) | (7) | (8) | (9) |
|---|---|---|---|---|---|---|---|---|---|
| | Heckman Two-Step | | 2SLS | | | Dependent Variables Measured by Different Methods | | | Independent Variable Replace |
| | First Stage | Second Stage | Lewbel | Mean | Lag | Non-Adjusted TSI | Adjusted TSI | Non-Adjusted DTSI | Direct Consumption Coefficient |
| SR | -0.076 | 1.811*** | 6.241** | 11.380*** | 13.015*** | 8.695*** | 5.711*** | 4.995*** | 2.392*** |
| | (0.06) | (0.16) | (2.95) | (1.53) | (1.26) | (1.08) | (1.03) | (1.01) | (0.45) |
| SRSQ | -0.797*** | -3.332*** | -14.806*** | -36.740*** | -30.912*** | -15.363*** | -12.646*** | -11.184*** | -4.027*** |
| | (0.13) | (0.31) | (2.94) | (3.15) | (1.67) | (2.41) | (1.96) | (1.94) | (0.65) |
| Intercept | 7.174*** | -2.819*** | | | | 8.511*** | 7.975*** | 7.660*** | 8.968*** |
| | (0.05) | (0.05) | | | | (1.18) | (1.31) | (1.28) | (1.16) |
| Control variables | Yes | Yes | Yes | Yes | Yes | Yes | Yes | Yes | Yes |
| IMR | | 0.251*** | | | | | | | |
| | | (0.02) | | | | | | | |
| rk LM test | | | 4157.00 | 1.3e+05 | 342.67 | | | | |
| P value | | | [0.000] | [0.000] | [0.000] | | | | |
| Wald rk F test | | | 2091.73 | 8.0e+04 | 171.49 | | | | |
| P value | | | [0.000] | [0.000] | [0.000] | | | | |
| Industry fixed effect | Yes | Yes | Yes | Yes | Yes | Yes | Yes | Yes | Yes |
| Province fixed effect | Yes | Yes | Yes | Yes | Yes | Yes | Yes | Yes | Yes |
| Year fixed effect | Yes | Yes | Yes | Yes | Yes | Yes | Yes | Yes | Yes |
| U-shaped test t value | | | | | | 3.682*** | 3.766*** | 3.291*** | 4.566*** |
| P value | | | | | | 0.000 | 0.000 | 0.001 | 0.000 |
| Inflection point | | | | | | 0.283 | 0.226 | 0.223 | 0.297 |
| Mean | | | | | | 0.236 | 0.236 | 0.236 | 0.228 |
| F statistics | | | 158.9 | 177.4 | 371.6 | 310.2 | 201.3 | 194.4 | 246.0 |
| Adjusted R-squared | | | | | | 0.125 | 0.082 | 0.079 | 0.123 |
| Observation | 989052 | 989052 | 989052 | 855787 | 855787 | 495207 | 989346 | 992412 | 992412 |

Note: *, ** and ** is statistically significant at 10%, 5% and 1%, respectively. The models controlled for industry, province and year fixed effects. The robust standard errors of clustering at city level are presented in parentheses. F statistic indicates that the model has joint significance.

variable and the industry mean value of the variable as the instrumental variables of the core independent variables [59, 60]. Referring to the above practice, this article takes the mean value of manufacturing service as its instrumental variable. Secondly, by using the Lewbel method [61, 62] for reference, this article constructs a new instrumental variable based on the heteroscedasticity identification method, which breaks through the limitation that traditional instrumental variable estimation must meet the exclusion constraints, and is more flexible in construction. It is not difficult to see from the estimation results that manufacturing servitization has an inverted U-shaped nonlinear effect on export technological sophistication, which is consistent with the benchmark result. Besides, the LM test values introduced by [63] passed the 1% level significance test, evidently rejecting the original hypothesis that the instrument variables were not recognized, indicating that the instrumental variables selected were identifiable. The Wald rk F test introduced by [63] passed the 1% level significance also passed the 1% level significance test, evidently rejecting the weak recognition hypothesis, indicating that the

tool variables selected have strong identifiability. The results of the above tests show that the instrumental variables constructed in this article based on the mean method and Lewbel method are effective.

**The test of the robustness of indicator.** First of all, the dependent variable in the benchmark regression is the adjusted export technological sophistication *DTSI* based on the export domestic value-added *DVA* of export and included in the product quality adjustment. For consideration of robustness, this article calculates the non-adjusted export technological sophistication *TSI* based on the export domestic value-added only, the adjusted export technological sophistication based on the exports and included in the product quality adjustment, and the non-adjusted export technological sophistication only based on the exports. Columns (6)-(7) report the relevant results, which are consistent with the benchmark results in terms of estimation coefficients and significance of core variables and control variables. It shows that the conclusion that manufacturing servitization has an inverted U-shaped nonlinear impact on export technological sophistication does not change with the change of measurement method of dependent variables.

Secondly, the core independent variable in the benchmark regression is the manufacturing servitization index measured based on the complete consumption coefficient matrix. For consideration of robustness, this article measures the manufacturing servitization index based on the direct consumption coefficient matrix. The results of column (8) show that manufacturing servitization has an inverted U-shaped impact on export technological sophistication, and the impact is in the promotion range. From what has been discussed above, the core results of this article do not change with the different methods of measuring the core independent variable.

## Expansibility analysis

### Based on the different sources of service input

Given the great difference like the service industry, and there are two sources of domestic and foreign service intermediate input, this article divides the manufacturing industry service intermediate input into consumption-oriented service intermediate input and production-oriented service intermediate input, and divides it into domestic and foreign types according to the source of service intermediate input. We further report the changing characteristics of manufacturing servitization from 2000 to 2010 in **Fig 5** based on the criteria of Retail, Transportation, Telecommunications, Finance, and R&D and the respective domestic and foreign input sources.

The regression results are presented in **Table 7**. Column (1)-(2) report the impact of manufacturing servitization of domestic and foreign service intermediate sources on export technological sophistication. The estimation coefficient of core variables and U-test results are 5% level, which shows that the manufacturing servitization of domestic and foreign service intermediate input sources has an inverted U-shaped impact on export technological sophistication. From the perspective of an inflection point and mean value, the impact is in the promotion range. In column (3)-(4), although the coefficients of manufacturing servitization and its square term of consumption-oriented service input sources are significant at 1% level, they fail to pass the U-shaped test, indicating that the manufacturing servitization of domestic consumption-oriented service input sources has a positive impact on export technological sophistication. In columns (5)-(6), the impact of manufacturing servitization of foreign production-oriented service input source does not pass the U-shaped test, which shows that manufacturing servitization of foreign production-oriented service input source has a positive impact on export technological sophistication. While manufacturing servitization of domestic production-oriented service input source has an inverted U-shaped impact on export technological

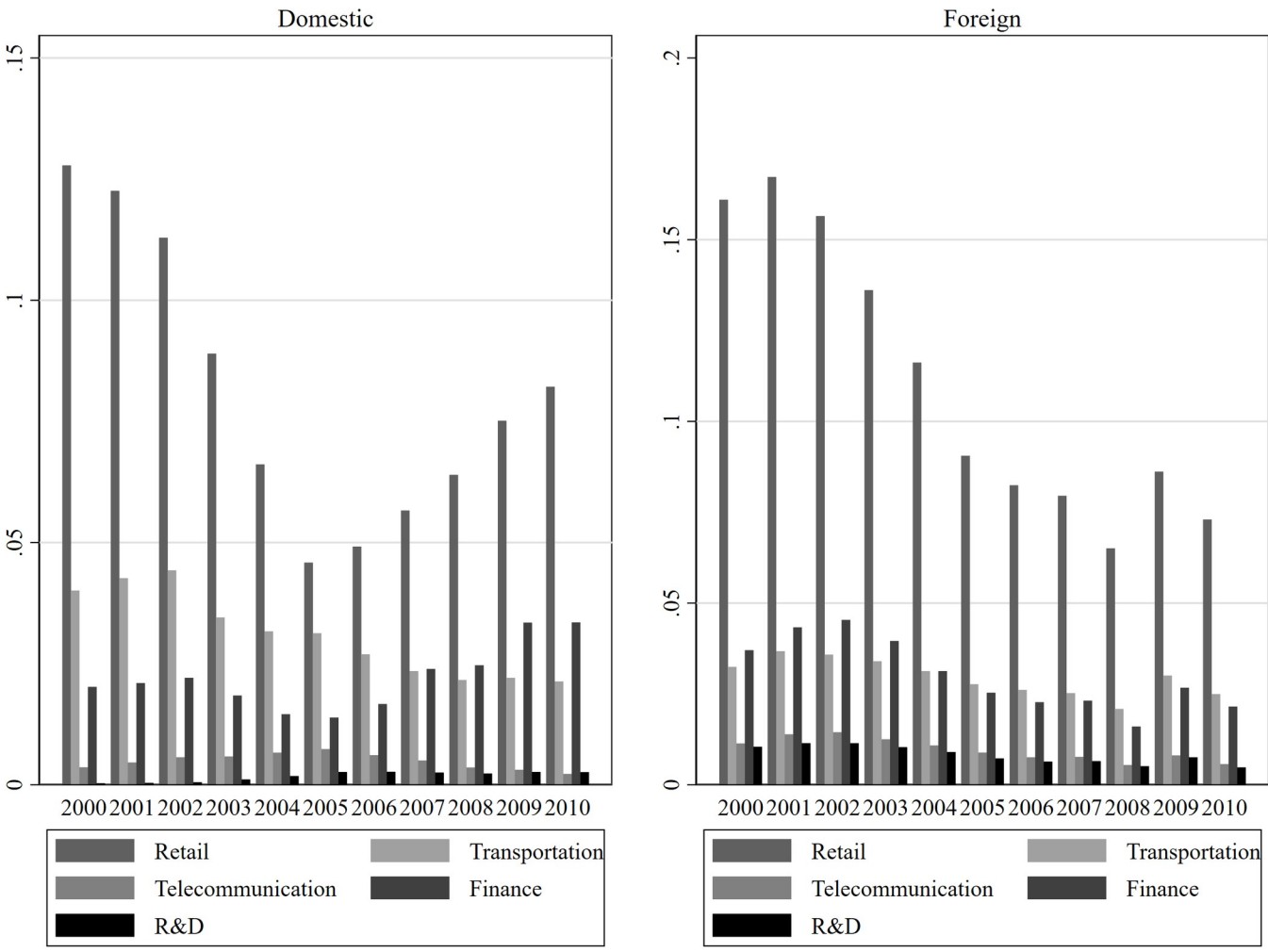

**Fig 5. The changing characteristics of manufacturing servitization in 2000–2010.**

sophistication, and it is in the promotion range. The reason for the difference in the impact of different sources of production-oriented service input on export technological sophistication may be that the domestic service input mostly has low value-added and less technology content, so it has a limited role in promoting the export technology content of domestic manufacturing enterprises. However, the service input provided by foreign countries is relatively high-end, which has a more obvious promotion impact on export technological sophistication.

There are differences in high-end factors such as human capital in different production-oriented services, such as the level of human capital in the financial industry, consulting and R&D service industry is significantly higher than that in the retail service industry. Therefore, the impact of production-oriented services with factor differentiation on the export technological sophistication of enterprises is also different. According to the classification of production-oriented services in WIOD, this article selects 5 sub-industries, namely retail industry (Retail), transportation industry (Transportation), telecommunication industry (Telecommunication), finance industry (Finance), and technical research and development industry (Technical & RD), to re-estimate the quantitative model, and the results are presented in **Table 8**.

**Table 7. The results of service input source (1).**

| | (1) | (2) | (3) | (4) | (5) | (6) |
|---|---|---|---|---|---|---|
| | Service Input | | Consumption-Oriented Service Input | | Production-Oriented Service Input | |
| Variable | Domestic | Foreign | Domestic | Foreign | Domestic | Foreign |
| SR | 3.781*** | 1.466* | 10.239*** | 0.357 | 4.333*** | 3.886*** |
| | (0.61) | (0.75) | (2.07) | (0.53) | (1.04) | (1.18) |
| SRSQ | -7.432*** | -2.010*** | -29.690*** | 8.722*** | -13.438*** | -8.024** |
| | (0.96) | (0.47) | (7.76) | (2.06) | (2.32) | (3.28) |
| Intercept | 8.847*** | 9.045*** | 8.531*** | 9.173*** | 8.971*** | 8.890*** |
| | (1.16) | (1.20) | (1.14) | (1.14) | (1.15) | (1.17) |
| Control variables | Yes | Yes | Yes | Yes | Yes | Yes |
| Industry fixed effect | Yes | Yes | Yes | Yes | Yes | Yes |
| Province fixed effect | Yes | Yes | Yes | Yes | Yes | Yes |
| Year fixed effect | Yes | Yes | Yes | Yes | Yes | Yes |
| U-shaped test t value | 4.859*** | 1.437* | | 1.574* | 3.180*** | 0.568 |
| P value | 0.000 | 0.076 | | 0.058 | 0.001 | 0.285 |
| Inflection point | 0.254 | 0.365 | 0.172 | -0.0205 | 0.161 | 0.242 |
| Mean | 0.208 | 0.339 | 0.115 | 0.0687 | 0.121 | 0.167 |
| F statistics | 281.8 | 253.1 | 279.0 | 247.2 | 265.2 | 236.8 |
| Adjusted R-squared | 0.119 | 0.123 | 0.120 | 0.121 | 0.123 | 0.123 |
| Observation | 989052 | 989052 | 989052 | 989052 | 989052 | 989052 |

Note: *, ** and ** is statistically significant at 10%, 5% and 1%, respectively. The models controlled for industry, province and year fixed effects. The robust standard errors of clustering at city level are presented in parentheses. F statistic indicates that the model has joint significance.

**Table 8. The results of service input source (2).**

| | (1) | (2) | (3) | (4) | (5) |
|---|---|---|---|---|---|
| Variable | Retail | Transportation | Telecommunication | Finance | Technical & RD |
| SR | 0.339 | 1.399*** | -0.019 | 0.556 | 0.364 |
| | (0.44) | (0.31) | (0.52) | (0.44) | (0.46) |
| SRSQ | -0.867*** | -1.542*** | -0.879*** | -1.122*** | -0.893*** |
| | (0.28) | (0.26) | (0.32) | (0.31) | (0.27) |
| Intercept | 9.276*** | 9.064*** | 9.359*** | 9.242*** | 9.275*** |
| | (1.12) | (1.14) | (1.16) | (1.15) | (1.16) |
| Control variables | Yes | Yes | Yes | Yes | Yes |
| Industry fixed effect | Yes | Yes | Yes | Yes | Yes |
| Province fixed effect | Yes | Yes | Yes | Yes | Yes |
| Year fixed effect | Yes | Yes | Yes | Yes | Yes |
| U-shaped test t value | 0.764 | 4.486*** | | 1.277 | 0.798 |
| P value | 0.223 | 0.000 | | 0.101 | 0.213 |
| Inflection point | 0.195 | 0.454 | -0.0107 | 0.248 | 0.204 |
| Mean | 0.222 | 0.242 | 0.239 | 0.240 | 0.255 |
| F statistics | 241.0 | 238.8 | 248.9 | 237.6 | 234.7 |
| Adjusted R-squared | 0.123 | 0.126 | 0.127 | 0.120 | 0.131 |
| Observation | 989052 | 989052 | 989052 | 989052 | 989052 |

Note: *, ** and ** is statistically significant at 10%, 5% and 1%, respectively. The models controlled for industry, province and year fixed effects. The robust standard errors of clustering at city level are presented in parentheses. F statistic indicates that the model has joint significance.

For the retail industry in column (1) and the telecommunication industry in column (3), the estimated coefficient of manufacturing servitization is not significant. For the technical research and development in column (5), the estimated coefficient of manufacturing servitization is not significant. For the finance industry in column (4), the estimated coefficient of manufacturing servitization is significant, while does not pass the U-shaped test. It indicates that the retail, telecommunication, technical research, and development service industry and other sources of manufacturing servitization do not have a non-linear relationship with export technological sophistication. For the transportation industry in column (2), the estimation coefficient of manufacturing servitization and U-shaped test have passed the 1% level significance test, indicating that the manufacturing servitization of the transportation industry has an inverted U-shaped impact on the export technological sophistication, and the impact is in the promotion range. The estimated coefficient of manufacturing servitization in columns (4) and column (5) is significant at a 1% level, indicating that the financial industry and technical R&D service industry have significantly promoted export technological sophistication. It is not difficult to understand the above conclusion that both the financial industry and the technical research and development service industry belong to the high-end service industry, which contains higher human capital, knowledge capital, and other high-level elements. Although the transportation industry belongs to the traditional labor-intensive or capital-intensive service industry like the retail industry and the telecommunications industry, it has the effect of "cost" due to the role of reducing transportation costs, thus promoting the improvement of export technological sophistication to a certain extent.

## The moderating effect of GVC participation

The existing research shows that manufacturing servitization plays an important role in promoting the international competitiveness of enterprises when the global value chain has deeply participated [64, 65]. However, there is still little literature on the economic effects of GVC (global value chain) participation based on the data of micro-enterprises, especially the further research on the combination of the new production division system of GVC and the servitization of the manufacturing industry. From the perspective of GVC participation, this paper examines whether it has a moderating effect on the impact of manufacturing servitization on export technological sophistication.

On the one hand, with the participation of the global value chain, enterprises can outsource the production segments which have no comparative advantage at a lower cost, which improves the production efficiency of enterprises. On the other hand, the increase of the degree of servitization of the manufacturing industry can improve the differentiation of products. When it has participated in the global value chain, it will highlight its competitive advantage.

The corresponding econometric model is specified as follows.

$$DTSI_{fit} = \alpha_0 + \beta_1(SR_{it} * GVCP_{jt}) + \beta_2(SRSQ_{it} * GVCP_{jt}) + \gamma X + \kappa_i + \mu_p + \eta_t + \varepsilon_{fit} \quad (11)$$

The related description is correspondent with the benchmark model in Eq (1). The interaction term of manufacturing servitization and GVC participation is the focus of the article. $GVCP_{jt}$ denotes the GVC participation at the industry level, which is measured based on the backward linkage method that followed the practice of [66]. It can be divided into three parts. Overall GVC participation index measures the overall participation of GVC of one industry, simple GVC participation index measures the participation of GVC that under the circumstance that intermediates flow through economies' boundaries one time, complex GVC participation index measures the participation of GVC that under the circumstance that intermediates flow through economies' boundaries more than one time.

**Table 9. The results of the moderating effect of GVC participation on the impact (1).**

| Variables | (1) | (2) | (3) | (4) | (5) | (6) | (7) | (8) | (9) |
|---|---|---|---|---|---|---|---|---|---|
| | Overall source servitization | | | Domestic source servitization | | | Foreign source servitization | | |
| | GVC participation | Simple GVC participation | Complex GVC participation | GVC participation | Simple GVC participation | Complex GVC participation | GVC participation | Simple GVC participation | Complex GVC participation |
| $SR^*GVCP$ | 7.662* | 12.037** | -0.864 | 10.479*** | 12.492*** | 2.194 | 6.077*** | 7.809*** | 2.401 |
| | (4.13) | (4.75) | (3.20) | (3.38) | (2.70) | (4.08) | (1.74) | (1.78) | (1.58) |
| $SRSQ^*GVCP$ | -9.194 | -28.386 | 8.602 | -22.231*** | -39.365*** | -2.133 | -7.456*** | -11.506*** | -4.923 |
| | (9.76) | (17.57) | (8.52) | (8.31) | (10.88) | (12.47) | (2.08) | (2.94) | (3.56) |
| Intercept | 9.040*** | 9.196*** | 9.246*** | 9.077*** | 9.241*** | 9.242*** | 9.079*** | 9.201*** | 9.261*** |
| | (1.10) | (1.13) | (1.11) | (1.11) | (1.13) | (1.11) | (1.08) | (1.12) | (1.11) |
| Control variables | Yes | Yes | Yes | Yes | Yes | Yes | Yes | Yes | Yes |
| Industry fixed effect | Yes | Yes | Yes | Yes | Yes | Yes | Yes | Yes | Yes |
| Province fixed effect | Yes | Yes | Yes | Yes | Yes | Yes | Yes | Yes | Yes |
| Year fixed effect | Yes | Yes | Yes | Yes | Yes | Yes | Yes | Yes | Yes |
| U-shaped test t value | | | 0.260 | | | | | | |
| P value | | | 0.398 | | | | | | |
| F statistics | 235.9 | 247.5 | 237.9 | 278.6 | 285.2 | 255.5 | 241.6 | 255.4 | 237.6 |
| Adjusted R-squared | 0.123 | 0.123 | 0.123 | 0.123 | 0.123 | 0.123 | 0.123 | 0.123 | 0.123 |
| Observation | 989052 | 989052 | 989052 | 989052 | 989052 | 989052 | 989052 | 989052 | 989052 |

Note: *, ** and ** is statistically significant at 10%, 5% and 1%, respectively. The models controlled for industry, province and year fixed effects. The robust standard errors of clustering at city level are presented in parentheses. F statistic indicates that the model has joint significance. In results of U-shaped test, extremums are outside interval in all columns except column (3), so the test is fail to reject the null hypothesis that the impact is monotone.

**Table 9** reported the results of a moderating effect of GVC participation. Column (1)-(3), column (4)-(6), and column (7)-(9) respectively report a moderating effect of GVC participation on the impact of manufacturing servitization on export technological sophistication. Take column (1)-(3) as an example, it can be seen from the estimated coefficients of interaction term that it is significantly positive in column (1)-(2), but not significant in column (3). The results show that the overall GVC participation level and simple GVC participation have a positive moderating effect on the correlation between manufacturing servitization and export technological sophistication, that is to say, GVC participation helps to promote the improvement of manufacturing servitization on the export technology level of enterprises. The possible explanation for the difference between the results of simple GVC and complex GVC is that the GVC participation based on backward linkage reflects the degree of connection between the specific industry and its upstream industry. The higher the simple GVC is, the more intermediate products of the industry subordinated to are in the upstream industry. Domestic enterprises use more of them to produce products with higher value-added exports. While complex GVC participation reflects the relationship between the specific industry and its upstream industry. The higher the GVC is, the more downstream industries the intermediate products belong to. Domestic enterprises are more likely to engage in low value-added businesses such as simple assembly and packaging, which is the "low-end locking" that participated in GVC. The results of other columns also get similar conclusions. Simple GVC has a positive

**Table 10. The results of the moderating effect of GVC participation on the impact (2).**

|  | (1) | (2) | (3) | (4) | (5) | (6) | (7) | (8) | (9) |
|---|---|---|---|---|---|---|---|---|---|
|  | Overall servitization | | | Domestic servitization | | | Foreign servitization | | |
| **Variables** | **25%** | **50%** | **75%** | **25%** | **50%** | **75%** | **25%** | **50%** | **75%** |
| SR*GVCP | -12.491 | 11.314* | 15.942*** | 9.631 | 23.345*** | 19.156*** | 10.909*** | 16.748*** | 12.884*** |
|  | (7.63) | (6.05) | (6.11) | (8.99) | (5.99) | (4.71) | (3.60) | (2.80) | (2.17) |
| SRSQ*GVCP | 71.770*** | -4.006 | -26.061 | -15.278 | -50.950*** | -43.200*** | -9.643* | -16.614*** | -12.597*** |
|  | (18.64) | (15.19) | (17.92) | (26.00) | (17.25) | (14.71) | (5.76) | (4.02) | (3.30) |
| Intercept | 3.127 | 6.991*** | 8.318*** | 3.057 | 6.980*** | 8.386*** | 2.805 | 6.777*** | 8.242*** |
|  | (3.03) | (1.71) | (1.38) | (3.02) | (1.71) | (1.40) | (3.02) | (1.71) | (1.37) |
| Control variables | Yes | Yes | Yes | Yes | Yes | Yes | Yes | Yes | Yes |
| Industry fixed effect | Yes | Yes | Yes | Yes | Yes | Yes | Yes | Yes | Yes |
| Province fixed effect | Yes | Yes | Yes | Yes | Yes | Yes | Yes | Yes | Yes |
| Year fixed effect | Yes | Yes | Yes | Yes | Yes | Yes | Yes | Yes | Yes |
| U-shaped test t value | 0.553 | | | | | | | | |
| P value | 0.290 | | | | | | | | |
| F statistics | 334.2 | 441.0 | 323.5 | 273.2 | 451.4 | 389.1 | 294.7 | 461.9 | 398.7 |
| Adjusted R-squared | 0.140 | 0.130 | 0.124 | 0.140 | 0.130 | 0.124 | 0.140 | 0.130 | 0.124 |
| Observation | 203663 | 432511 | 614163 | 203663 | 432511 | 614163 | 203663 | 432511 | 614163 |

Note: *, ** and ** is statistically significant at 10%, 5% and 1%, respectively. The models controlled for industry, province and year fixed effects. The robust standard errors of clustering at city level are presented in parentheses. F statistic indicates that the model has joint significance. In results of U-shaped test, extremums are outside interval in all columns except column (1), so the test is fail to reject the null hypothesis that the impact is monotone.

moderating effect on the correlation between manufacturing servitization with domestic and foreign sources of service input, and export technological sophistication.

Because the outsourcing of manufacturing services (including offshore outsourcing and onshore outsourcing) requires a lot of sunk costs input, many enterprises with higher production efficiency adopt outsourcing strategy, which increases the market share of high-efficiency enterprises and reduces the market share of low-efficiency enterprises, resulting in the reallocation of production factors among enterprises with different efficiency [67]. According to the reallocation effect, enterprises with different efficiency have different feedback on the positive incentive of manufacturing servitization, in other words, there is a heterogeneity on the moderating effect of manufacturing servitization. Therefore, this article divides the enterprises into three different subsamples according to the level of efficiency, which are the top 25%, 50%, and 75% of the efficiency level, to examine the in using quantile regression.

**Table 10** reports the quantile regression results of the moderating effect of GVC participation. From the estimated coefficient of core variables, a square term of the interaction is not significant or fails to pass the U-test, which indicates that the moderating effect of GVC participation is monotonic. It can be seen from columns (1)-(3) and (4)-(6) that GVC participation has a more positive moderating effect on enterprises with low and medium productivity. The results show that when GVC participation is high, manufacturing servitization significantly promotes the export technological sophistication of low-efficiency enterprises, which is similar to previous literature conclusions [65, 68]. The possible explanation is that there are a large number of processing trade enterprises engaged in low value-added business in China, and their productivity can not accurately reflect their international competitiveness, that is, there is a "productivity paradox" phenomenon. In column (7), GVC participation has a significant positive moderating effect on manufacturing servitization, and the export technological

sophistication of high-efficiency enterprises has a significant positive moderating effect, which is related to the service input with more high-end elements from abroad used by enterprises.

## Conclusion and discussions

This article investigates the impact of manufacturing servitization on the export technological sophistication of enterprises in China during 2000–2010. Firstly, based on the perspective of value-added of trade, this article estimates the servitization of the manufacturing industry and export technological sophistication of enterprises and tests the impact of the manufacturing servitization on the export technological sophistication of enterprises. The main conclusions are as follows. First, the manufacturing service industry as a whole has an inverted U-shaped impact on the export technological sophistication of enterprises, which is in the promotion range. Second, from the perspective of enterprise characteristics and industry characteristics, manufacturing servitization has an inverted U-shaped impact on export technological sophistication of mixed trade enterprises, central-western and eastern enterprises, domestic and foreign enterprises, and those impacts are in the promotion range. Manufacturing servitization has an inverted U-shaped impact on the export technological sophistication of knowledge-intensive industries, and the impact is in the promotion range. Third, from the perspective of service input sources, manufacturing servitization from domestic and foreign service input sources has an inverted U-shaped impact on export technological sophistication, and the impact is in the promotion range. Manufacturing servitization from domestic consumption-oriented service input sources and foreign production-oriented service input sources has a promoting impact while manufacturing servitization from domestic production-oriented service input sources has an inverted U-shaped impact on export technological sophistication, and it is in the promotion range. Manufacturing servitization of the financial industry and technical research and development service industry has a promotion effect while manufacturing servitization of the transportation industry has an inverted U-shaped effect, which is in the promotion range. Fourth, overall GVC participation level and simple GVC participation have a positive moderating effect on the correlation between manufacturing servitization and export technological sophistication. When GVC participation is high, manufacturing servitization significantly promotes the export technological sophistication of low-efficiency enterprises.

This study enriches the research field of manufacturing servitization and export technological sophistication, deepens the understanding of the relationship between the two, and strengthens the theoretical foundation for the transmission mechanism of the impact from the perspective of the "spillover" and "cost" effects. Besides, the conclusions of this article have important implications for decision-makers to enhance the international competitiveness of enterprises. First, impact heterogeneity findings have practical implications. The results show that, in the process of promoting manufacturing servitization, decision-makers should fully consider the factors such as trade pattern, region and ownership, and industrial differentiation, gradually improve the development of modern service industry in the central-western regions, promote the expansion of processing trade enterprises towards the front and back ends of the industrial chain. Encourage and guide manufacturing enterprises to continuously integrating high-end service elements into R&D innovation, market research, after-sales service, and other aspects. Second, the conclusions of the mechanism test have practical implications. The research on mediating effect shows that R&D innovation and cost reduction are important channels for manufacturing servitization to promote export technological sophistication of enterprises. Therefore, decision-makers should continue to implement the central government's major goals of supply-side structural reform and cost reduction in the "Three

Deduction, One Reduction, One Compensation" strategy, adapt to and lead the new normal economy through innovation-driven. Promote industrial optimization and upgrading with the conversion of new and old kinetic energy. Finally, the findings regarding the sources of service factor inputs are equally relevant in practice. From the conclusion of the research on the difference of service input sources, we can see that China's manufacturing industry is at the middle and low end of the global value chain, resulting in less demand for high-end services such as local research and development, technical services and financial services. Therefore, we should speed up the opening up of service industries such as finance, technical consultation, and research and development, and skills training, implement policies such as taxation, finance to support manufacturing servitization. Promote the positive role of manufacturing servitization in the international competitiveness of enterprises by strengthening the policy guarantee of capital, human capital, and other aspects of manufacturing servitization, to achieve high-quality development of China's foreign trade.

Finally, it is necessary to state that there are also some limitations in this article's study. First, we confirm the U-shaped characteristic of the impact by introducing a quadratic term of the variable, which is closer to the truth than previous linear examinations. However, the existence of more complex impact characteristics needs to be confirmed by subsequent studies. Second, among the many dimensions of heterogeneity characteristics, we only examined three of them, such as firm trade mode, location, and ownership, but heterogeneity characteristics such as firm size, firm age, and firm debt level were not well observed and need to be further examined. Third, we confirm the important role played by R&D and cost factors in the transmission mechanism, but other factors such as industrial agglomeration, FDI, and external shocks such as industrial policy are not examined. The above questions may be answered in future studies.

## Author Contributions

**Conceptualization:** Yuanhong Hu, Yixin Dai.

**Data curation:** Yuanhong Hu, Yixin Dai.

**Formal analysis:** Yuanhong Hu.

**Methodology:** Yuanhong Hu.

**Software:** Yuanhong Hu.

**Writing – original draft:** Yuanhong Hu.

**Writing – review & editing:** Sheng Sun, Min Jiang.

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
