## [Decision Letter · Decision Letter 0]

17 Mar 2021

PONE-D-20-40809

Research on the Promoting Effect of Servitization on Export Technological Sophistication of Manufacturing Enterprises

PLOS ONE

Dear Dr. Dai,

Thank you for submitting your manuscript to PLOS ONE. After careful consideration, we feel that it has merit but does not fully meet PLOS ONE’s publication criteria as it currently stands. Therefore, we invite you to submit a revised version of the manuscript that addresses the points raised during the review process.

We look forward to receiving your revised manuscript.

Kind regards,

Jintao Lu

Academic Editor

PLOS ONE

Journal Requirements:

2. Please include your tables as part of your main manuscript and remove the individual files. Please note that supplementary tables should be uploaded as separate "supporting information" files.

We note that one or more of the authors are employed by a commercial company: Postal Savings Bank of China.

4.1. Please provide an amended Funding Statement declaring this commercial affiliation, as well as a statement regarding the Role of Funders in your study. If the funding organization did not play a role in the study design, data collection and analysis, decision to publish, or preparation of the manuscript and only provided financial support in the form of authors' salaries and/or research materials, please review your statements relating to the author contributions, and ensure you have specifically and accurately indicated the role(s) that these authors had in your study. You can update author roles in the Author Contributions section of the online submission form.

4.2. Please also provide an updated Competing Interests Statement declaring this commercial affiliation along with any other relevant declarations relating to employment, consultancy, patents, products in development, or marketed products, etc.  

Reviewers' comments:

Reviewer's Responses to Questions

**Comments to the Author**

1. Is the manuscript technically sound, and do the data support the conclusions?

Reviewer #1: Yes

Reviewer #2: No

2. Has the statistical analysis been performed appropriately and rigorously? 

Reviewer #1: Yes

Reviewer #2: No

3. Have the authors made all data underlying the findings in their manuscript fully available?

Reviewer #1: Yes

Reviewer #2: No

4. Is the manuscript presented in an intelligible fashion and written in standard English?

Reviewer #1: Yes

Reviewer #2: No

5. Review Comments to the Author

Reviewer #1: Introduction

Research on the Promoting Effect of Servitization on Export Technological Sophistication of Manufacturing Enterprises; starting from the introduction, the section is well written. In the current version, the paragraph is strong: it does provide enough information about the research topic, the reference context, and the gap that the work wants to fill. The function of the introduction is to describe the reference context from which the research question originates and to highlight the research gap to give evidence of the contribution of the paper. In the actual version of the paragraph, this logical sequence is well introduced and adequately supported. So, the entire section is well written, highlighting the relevance of the theme, the gap in literature and the establishment of consequent research aims.

Theoretical framework and literature review

In general terms, the paper has fundamental theoretical framework and literature review, started with background and hypothesis building. And indeed, there is evidence of organization and clarity.

Hypotheses

In general, all proposed hypothesis is adequately supported. All the hypotheses are supported by the references linking it to the author's work objectives. The moderating effect of GVC participation s well described and adequate.

Methodology and Design

The explanation of measurement variables is done in good length and precise. Adequate data analyses to arrive at the results indicating robustness of the analyses.

Conclusion

The paragraph conclusion is adequate in the current form.

In the paper, there is no section dedicated to the theoretical and practical implications. More details are needed about (a) research limitations, and (b) the managerial and theoretical implications. Theoretical and managerial implications could be more extensively debated consistent with the results obtained.

Reviewer #2: - 3. Research design

What is "econometrical"? Please write its definition.

The description of your econometrical model is poor.

-- 3.1 Econometrical model specification

Please write all definitions of the variables in Equation (1).

What is alpha, beta, ....?

"Based on existing literature, this article selects the following variables." What is the "existing literature"?

- 4. Empirical results and analysis

NO Figures!

Please show and summarize your result visually to improve PLOS ONE readers' understanding.

6. PLOS authors have the option to publish the peer review history of their article (what does this mean?). If published, this will include your full peer review and any attached files.

Reviewer #1: **Yes: **Anantha Raj A. Arokiasamy

Reviewer #2: No

---

## [Author Response · Author response to Decision Letter 0]

9 May 2021

Response to Reviewers

Response to Reviewer #1:

Thank you for your helpful suggestions on this article

In the Conclusions and Discussions part, we have adopted your suggestion and added the content of the study limitations. The theoretical and practical implications of the study were mentioned in the original paper, and before resubmission, we further clarified them and improved the coherence with the obtained conclusions.

Response to Reviewer #2:

Thank you for your helpful suggestions on this article.

Response to - 3. Research design

We provide detailed definitions and explanations of econometric concepts to help readers understand the disciplinary basis of our modeling, "Econometrics is a discipline that uses mathematical and statistical methods to determine specific quantitative relationships in economic relationships. Econometrics measures and validates actual statistics of economic relationships and provides quantitative information for the qualitative account of economic theory on the dependence of economic variables to predict the future and provide a scientific basis for economic planning and determination of economic policy. In the subsequent study, we construct models based on econometric principles."

Response to -- 3.1 Econometrical model specification

Due to an oversight, we did not have an explanation of these parameters in the original draft, but we have now added an explanation of this section, "For estimated parameters, denotes the intercept term, or constant term is used to fit the model and has no practical significance. and denote the estimated parameters of the primary and secondary terms of the core variable manufacturing servitization, respectively. denotes estimated parameters representing the set of control variables. For the estimated parameters of the fixed effects, , and denote the industry level, provincial level, and year level fixed effects and random errors, respectively. denotes the random perturbation terms."

Besides, we add specific literature to clarify the reference base for control variables.

Response to - 4. Empirical results and analysis

At the request of the reviewers, we added Figures in several parts of the article to facilitate the reader's understanding of the study through visualization. In the Indicator measure part, the Mediation effect analysis part, the Heterogeneity results and analysis part, and the Expansibility analysis part, a figure was added respectively.

---

## [Decision Letter · Decision Letter 1]

11 Jun 2021

PONE-D-20-40809R1

Research on the Promoting Effect of Servitization on Export Technological Sophistication of Manufacturing Enterprises

PLOS ONE

Dear Dr. Dai,

Thank you for submitting your manuscript to PLOS ONE. After careful consideration, we feel that it has merit but does not fully meet PLOS ONE’s publication criteria as it currently stands. Therefore, we invite you to submit a revised version of the manuscript that addresses the points raised during the review process.

We look forward to receiving your revised manuscript.

Kind regards,

Jintao Lu

Academic Editor

PLOS ONE

Journal Requirements:

Reviewers' comments:

Reviewer's Responses to Questions

**Comments to the Author**

1. If the authors have adequately addressed your comments raised in a previous round of review and you feel that this manuscript is now acceptable for publication, you may indicate that here to bypass the “Comments to the Author” section, enter your conflict of interest statement in the “Confidential to Editor” section, and submit your "Accept" recommendation.

Reviewer #1: All comments have been addressed

Reviewer #3: All comments have been addressed

2. Is the manuscript technically sound, and do the data support the conclusions?

Reviewer #1: Yes

Reviewer #3: Yes

3. Has the statistical analysis been performed appropriately and rigorously? 

Reviewer #1: Yes

Reviewer #3: Yes

4. Have the authors made all data underlying the findings in their manuscript fully available?

Reviewer #1: Yes

Reviewer #3: Yes

5. Is the manuscript presented in an intelligible fashion and written in standard English?

Reviewer #1: Yes

Reviewer #3: Yes

6. Review Comments to the Author

Reviewer #1: The authors have addressed all the comments of the reviewers and revised the manuscript

accordingly. They have provided the detailed descriptions on the structure and the procedure of the gap estimation, which now strengthen the experimental results.

From my point of view, the paper is almost suitable for publication in PLOS ONE.

I recommend accepting of this manuscript.

Reviewer #3: This article comprehensively applied econometric models to study the impact of China's manufacturing servitization on export technological sophistication from 2000 to 2010.

There are some little problems that should be improved:

The full name of the abbreviations should be given when it is used at the first time, such as GVC, HS.

“With the comparative advantage of low production costs, China has gained the rapid growth of exports.” Can you provide a reference for this sentence?

Can you provide a more detailed illustration for the computation of HHIit, and provide short words to show the meaning of DTSIfit.

In the last paragraph of section “Econometrical model specification”, “For the estimated parameters of the fixed effects……”， the number of subject and object are not consistent.

Can you supplement the information on the software and tools which was used to construct your models (mediation effect analysis, Heckman two-step method) and in your statistical works in your manuscript? Such as version. More information (code) can be freely accessed?

“Observation” in the header of Table 1 means the number of entries? Then the procedures used for missing variables can be shown? Ref. 35 has been downloaded, but I did not find out which steps were adopted by them to preprocess missing indicators.

Please provide the concrete definition or reference for “adjusted enterprise”

7. PLOS authors have the option to publish the peer review history of their article (what does this mean?). If published, this will include your full peer review and any attached files.

Reviewer #1: **Yes: **Anantha Raj A. Arokiasamy

Reviewer #3: **Yes: **Jie Zhang

---

## [Author Response · Author response to Decision Letter 1]

6 Jul 2021

Response to the Academic Editor

Dear editor,

We checked the reference list and confirmed its correctness and completeness. In addition, we added two additional publications due to the reviewer's suggestion and marked them in red in the list.

Response to Reviewers

Dear Reviewer #3,

Thank you for your constructive suggestions for the improvement of the manuscript. According to your suggestions and some of your questions, we reply to them one by one.

Q1: The full name of the abbreviations should be given when it is used at the first time, such as GVC, HS.

R1: We show the first appearance of a term by its full name. (P1, P7) 

Q2: "With the comparative advantage of low production costs, China has gained the rapid growth of exports." Can you provide a reference for this sentence?

R2: We provide a literature basis for this sentence. (P2)

Q3: Can you provide a more detailed illustration for the computation of HHIit, and provide short words to show the meaning of DTSIfit.

R3: We provide a more detailed explication of the dependent variable DTSI for the model. denotes export technological sophistication of enterprises, as one of the important indicators of export performance, it measures the level of export product quality of enterprises in a narrow sense and the export competitiveness of enterprises in a broad sense (P5). Besides, we provide a more detailed illustration for the computation of HHI. A Herfindal index formula. In this formula, the sum of the squares of the share of sales of each industry in the total sales of a region constitutes the degree of industrial agglomeration in that region. When the proportion of a certain industry is larger, the final value will also be larger. (P6).

Q4: In the last paragraph of section “Econometrical model specification”, “For the estimated parameters of the fixed effects……”， the number of subject and object are not consistent.

R4: The original sentence was incorrectly translated before, and we correct it. We thank the reviewers for pointing out this error. (P6)

Q5: Can you supplement the information on the software and tools which was used to construct your models (mediation effect analysis, Heckman two-step method) and in your statistical works in your manuscript? Such as version. More information (code) can be freely accessed?

R5: Throughout the manuscript, we use the statistical software StataMP16.1 to measure , and other control variables. In addition, we use this software to perform panel fixed effects regressions on the models constructed in the whole manuscript, including mediating effects regressions to discriminate the impact mechanism and Heckman-Two Step method regressions to discriminate endogeneity (P9). The availability of the code is not publicly available because it is still a part of our phase study, but we provide detailed information on the steps for the metric measurement and model regression in the manuscript.

Q6: "Observation" in the header of Table 1 means the number of entries? Then the procedures used for missing variables can be shown? Ref. 35 has been downloaded, but I did not find out which steps were adopted by them to preprocess missing indicators.

R6: "Observation" that appears in each table indicates the entry used for the study. Because the presence of missing values for each entry is not allowed in the panel regression, the statistical software Stata automatically eliminates entries with missing values from the model during the regression process, so the number of entries presented in Table 1 does not necessarily equal the number of entries in the subsequent tables of the regression results (P9). Besides, the original reference was cited incorrectly and we have corrected it, for which we are grateful. In fact, we fully refer to Brandt et al. (2012) and Feenstra et al. (2014). However, it should be noted that in modeling regressions, the statistical software automatically eliminates a small number of entries that may lead to covariance due to the introduction of multidimensional fixed effects (P8).

Q7: Please provide the concrete definition or reference for "adjusted enterprise".

R7: In the manuscript, the term "adjusted enterprise" is intended to refer to export technological sophistication at the enterprise level after the product quality has been corrected, not to the enterprise itself. (P7)

---

## [Decision Letter · Decision Letter 2]

27 Jul 2021

Research on the Promoting Effect of Servitization on Export Technological Sophistication of Manufacturing Enterprises

PONE-D-20-40809R2

Dear Dr. Dai,

We’re pleased to inform you that your manuscript has been judged scientifically suitable for publication and will be formally accepted for publication once it meets all outstanding technical requirements.

Kind regards,

Jintao Lu

Academic Editor

PLOS ONE

Additional Editor Comments (optional):

Reviewers' comments:

Reviewer's Responses to Questions

**Comments to the Author**

1. If the authors have adequately addressed your comments raised in a previous round of review and you feel that this manuscript is now acceptable for publication, you may indicate that here to bypass the “Comments to the Author” section, enter your conflict of interest statement in the “Confidential to Editor” section, and submit your "Accept" recommendation.

Reviewer #3: All comments have been addressed

2. Is the manuscript technically sound, and do the data support the conclusions?

Reviewer #3: Yes

3. Has the statistical analysis been performed appropriately and rigorously? 

Reviewer #3: Yes

4. Have the authors made all data underlying the findings in their manuscript fully available?

Reviewer #3: Yes

5. Is the manuscript presented in an intelligible fashion and written in standard English?

Reviewer #3: Yes

6. Review Comments to the Author

Reviewer #3: This study investigates the impact of manufacturing servitization on the export technological sophistication of enterprises in China during 2000-2010. It has complete data analyses, including benchmark results and analysis, mediation effect analysis, heterogeneity results and analysis, and the robustness test., and impressive results.

7. PLOS authors have the option to publish the peer review history of their article (what does this mean?). If published, this will include your full peer review and any attached files.

Reviewer #3: **Yes: **Jie Zhang

---

## [Editor Report · Acceptance letter]

19 Aug 2021

PONE-D-20-40809R2 

Research on the Promoting Effect of Servitization on Export Technological Sophistication of Manufacturing Enterprises 

Dear Dr. Dai:

I'm pleased to inform you that your manuscript has been deemed suitable for publication in PLOS ONE. Congratulations! Your manuscript is now with our production department. 

Kind regards, 

on behalf of

Dr. Jintao Lu 

Academic Editor

PLOS ONE